# HYPOLE: Hyperproperty-Guided Multi-Agent Reinforcement Learning under Partial Observation

**Arshia Rafieioskouei** [1]  **Tzu-Han Hsu** [1]  **Matthew Lucas** [1]  **Borzoo Bonakdarpour** [1]

## Abstract

Formal specification is a powerful tool to guide the learning process and provides significant advantages over reward shaping: (1) mathematical rigor; (2) expressiveness to specify objectives and constraints, and (3) the ability to define tactics to achieve objectives. However, these benefits remain largely unexplored in the context of Multi-Agent Reinforcement Learning (MARL). This paper introduces HYPOLE, a novel framework for MARL under *partial observability*, where learning is guided by the expressive power of the so-called *hyperproperties* and, in particular, the temporal logic HyperLTL. We integrate Centralized Training for Decentralized Execution (CTDE) techniques with HYPOLE to synthesize *decentralized* policies, and our evaluation on SMAC, MessySMAC, and WildFire benchmark demonstrates clear advantages over baselines.

## 1. Introduction

Multi-Agent Reinforcement Learning (MARL) is challenging due to combinatorial scaling, and becomes even more demanding when agents must satisfy multiple interdependent objectives and constraints. Existing MARL techniques (Rashid et al., 2018; Sunehag et al., 2017; Son et al., 2019) are strangers to *symbolic* techniques that provide a wealth of powerful specification languages that express complex objectives and constraints among agents and guide the learning process. Moreover, shaped reward functions are often insufficient for expressing and guiding learning with respect to *relational* objectives among agents and their temporal behaviors (Alur et al., 2026). More importantly, a particular limitation of existing techniques is that shaped rewards for a set of objectives are implicitly *universally quantified*; that is, the intended objective must hold for every pair of agents' behaviors. This inherent universal strictness results in limiting the search space for better policies.

To explain the issue of universality, consider the following formula in predicate logic: $\varphi_1 = \forall x.\forall y.(x < y)$, where $x$ and $y$ range over integers. This formula is obviously false, but a weaker variation of this formula $\varphi_2 = \forall x.\exists y.(x < y)$ is valid. In fact, synthesizing a policy to select $y$ in response to any choice of $x$ is to learn a witnessing *Skolem function* (Skolem, 1879) such as $y = \mathbf{f}(x) = x + 1$. In the world of MARL, virtually all existing approaches attempt to search for policies for objectives that are of the form $\forall\forall.\psi$, while many multi-agent requirements are of the form $\forall\exists.\psi$, expressing that for every behavior of one agent, there exists a behavior of another agent, such that the objective is satisfied. For example, (Hsu et al., 2025; Beutner & Finkbeiner, 2024) formulate multi-agent planning problems, using $\forall\exists.\psi$ specifications. Moreover, (Hsu et al., 2025) extends beyond multi-agent planning by encoding the single-agent planning problem DeepSea Treasure (Vamplew et al., 2011) as a multi-agent problem under a $\forall\exists.\psi$ specification.

We propose a specification-guided method that boosts MARL under *partial observability*. We use the concept of *hyperproperties* and, in particular, the temporal logic HyperLTL (Clarkson et al., 2014) as our specification language to express the intended behavior of each agent as well as their *relational* (inter-dependent) objectives and constraints. Our proposed method HYPOLE (**Hy**perproperties for **P**artially **O**bservable **L**earning **E**nvironments), is built on realistic multi-agent settings, assuming that agents operate in an environment modeled as a partially observable Markov decision process (POMDP) with the goal of learning a *decentralized* set of policies, one per agent (see Fig. 1).

**Our Contributions.** (1) We formulate the policy synthesis problem for a multi-agent system as a learning problem, where the goal is to maximize the probability of satisfying a HyperLTL specification that expresses a set of objectives and constraints under partial observability (Sec. 4). (2) We solve this problem by first transforming symbolic satisfaction of the HyperLTL specification for a POMDP into an optimization problem by defining continuous robustness

---

[1]Department of Computer Science and Engineering, Michigan State University, MI, USA. Correspondence to: Arshia Rafieioskouei <rafieios@msu.edu>, Borzoo Bonakdarpour <borzoo@msu.edu>.

*Proceedings of the 43rd International Conference on Machine Learning*, Seoul, South Korea. PMLR 306, 2026. Copyright 2026 by the author(s).

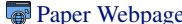
Paper Webpage

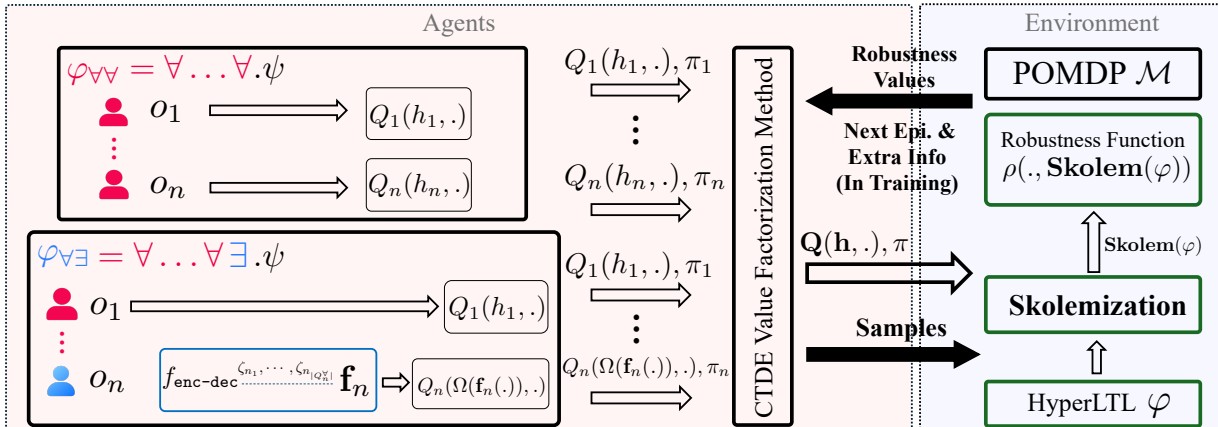

**Figure 1.** Overview of HYPOLE.

functions, then use off-the-shelf Centralized Training for Decentralized Execution (CTDE) algorithms to learn an optimal collection of decentralized policies (Sec. 5), and (3) we evaluate HYPOLE on scenarios from (i) the StarCraft II Multi-Agent Challenge (SMAC) (Samvelyan et al., 2019), (ii) MessySMAC (Phan et al., 2023), and (iii) the WildFire benchmark. Our results show that HYPOLE coupled with CTDE algorithms is more efficient and effective than the vanilla versions in handling complex requirements. We also show how HyperLTL assists in (1) faster learning of better policies, and (2) elegantly express various tactics (Sec. 6).

## 2. Related Work

We believe that the work closest to HYPOLE is HYPRL (Hsu et al., 2025), which studies hyperproperty-guided MARL in a fully observable and centralized setting. Both approaches leverage Skolemization, a well-known technique in logic for handling quantifier alternations. However, HYPOLE introduces a new challenge from the decentralized setting: the Skolem function does not have access to the traces of other agents when reasoning about its own behavior relative to theirs. Moreover, because HYPRL operates in a centralized setting, it does not scale well with increasing environment complexity or the number of agents. In contrast, HYPOLE addresses this scalability challenge by learning decentralized policies in partially observable multi-agent environments. Other work on specification-guided RL include (Jothimurugan et al., 2021; Li et al., 2017; Kuo et al., 2020; De Giacomo et al., 2019; Hasanbeig et al., 2019; Xu & Topcu, 2019; Yuan et al., 2019; Le et al., 2024). However, these approaches are either limited to single-agent tasks or fail to scale effectively in MARL settings due to the composition of MDPs. The extensions to multi-agent systems (Jothimurugan et al., 2022; ElSayed-Aly & Feng, 2022; Liu et al., 2024; León & Belardinelli, 2020; Hammond et al., 2021) either rely on centralized

constructions, employ enumerate-and-verify strategies, or support only co-safety properties. Several work integrate temporal logic constraints into POMDP planning, including finite-state controllers and barrier-based methods for LTL and DTL specifications (Sharan & Burdick, 2014; Ahmadi et al., 2020; Wang et al., 2021; Liu et al., 2021), but typically focus on single-agent or small settings. Furthermore, Inverse RL (Choi & Kim, 2011) recovers rewards from expert demonstrations, whereas HYPOLE uses only a hyperproperty specification and assumes no expert behavior.

## 3. Preliminaries

Let $\Delta(X) \triangleq \left\{ p : X \mapsto [0,1] \mid \sum_{x \in X} p(x) = 1 \right\}$ be the probability simplex of a set $X$ of random variables. We abbreviate tuples $\langle x_1, \ldots, x_n \rangle$ as $\langle x_i \rangle_{i \in \{1,\ldots,n\}}$.

### 3.1. Partially Observable Markov decision Processes

**Definition 3.1.** A *partially observable Markov decision process (POMDP)* is a tuple $\mathcal{M} = \langle S, I, A, P, R, \mathcal{O}, O, \mathsf{AP}, L, \gamma \rangle$, where $S$ is a finite set of *states*, $I$ is *initial* distribution over $S$, i.e., $I \in \Delta(S)$, $A$ is a finite *action* space, the transition function $P(s' \mid s, a)$ specifies the conditional probability of transitioning to $s' \in S$ after taking action $a \in A$ in $s \in S$. By taking $a$ in $s$, the agent receives a scalar *reward* $R(s, a)$ and $\gamma \in (0, 1]$ is discount factor. An *observation* $o \in O$ is produced by observation probability function $\mathcal{O}(o \mid s, a)$ given the current $s$ and $a$. AP denotes a finite set of *atomic propositions*, and $L : S \to 2^{\mathsf{AP}}$ is a *labeling* function. □

**Example.** Consider the environment in Fig. 2, a Wild-Fire scenario on a $3 \times 3$ grid world with cells labeled $\{a, b, \ldots, i\}$. A firefighter drone (FF 🚁) is tasked to extinguish three fire zones 🔥, and a medical drone (Med 🚁) whose objective is to rescue two victims 🧍. Agents can communicate only within a distance of three. Med is not

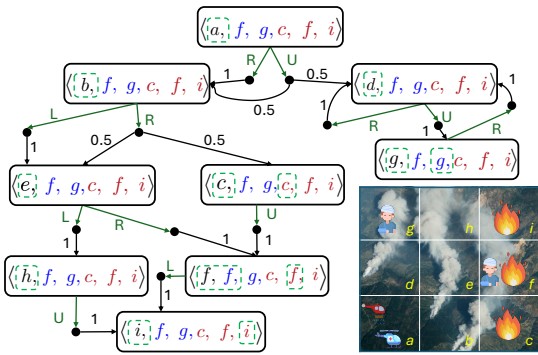

**Figure 2.** WildFire scenario and its POMDP.

allowed to enter fire-zone. The map is partially observable due to smoke and the agents can detect fire or victims only when they are in the same cell. The underlying state consists of the agents' positions and the locations of the two victims and three fire zones. Fig. 2 illustrates a POMDP, with state $\langle \texttt{pos}, \texttt{V1}, \texttt{V2}, \texttt{F1}, \texttt{F2}, \texttt{F3} \rangle$ (agent position, victim positions, and fire positions). Observations are partial; e.g., in state $\langle a, f, g, c, f, i \rangle$ the agent's observation is $\langle a \rangle$.

A *history* is an observation-action sequence $h \triangleq o_0 \xrightarrow{a_0} o_1 \xrightarrow{a_1} \cdots$, where for all $i \geq 0$, $o_i \in O$ and $a_i \in A$. A *sub-history* $h_{[\ell:k]}$ is a finite segment $o_l \xrightarrow{a_l} \cdots \xrightarrow{a_{k-1}} o_k$, for $0 \leq \ell < k < |h|$. We denote the set of all finite histories by $\mathcal{H}^*$. A *policy* $\pi : \mathcal{H}^* \to A$ maps a history to a fixed action. A *path* in $\mathcal{M}$ is a sequence of states $\zeta \triangleq s_0 \xrightarrow{a_0} s_1 \xrightarrow{a_1} \cdots$, where $s_i \in S$ and $a_i \in A$ for all $i \geq 0$. A *sub-path* $\zeta_{[\ell:k]}$ is a finite segment $s_\ell \xrightarrow{a_\ell} \cdots \xrightarrow{a_{k-1}} s_k$, for $0 \leq \ell < k < |\zeta|$. Given a path $\zeta$, the *trace* $t$ of $\zeta$ is a sequence of labels $\mathsf{Tr}(\zeta) \triangleq t(0)t(1)\cdots$, such that $t(i) = L(s_i)$ for all $i \geq 0$. We write $\mathcal{Z}^*$ to denote the set of all finite paths, and $\mathsf{Traces}(\mathcal{Z}^*)$ to denote the set of all finite traces of $\mathcal{M}$.

### 3.2. Finite Semantics for HyperLTL

HyperLTL syntax (Clarkson et al., 2014) is as follows:

$$\varphi ::= \exists \tau. \varphi \mid \forall \tau. \varphi \mid \psi \qquad \psi ::= p_\tau \mid \neg \psi \mid \psi \vee \psi \mid \bigcirc \psi \mid \psi \, \mathcal{U} \, \psi,$$

where $p \in \mathsf{AP}$ is an atomic proposition and $\tau$ is a trace variable. The second rule yields standard LTL formulas, with Boolean connectives $\neg$, $\vee$, and temporal operators $\bigcirc$ ('next') and $\mathcal{U}$ ('until'). Other Boolean and temporal operators are defined as syntactic sugar: true $\triangleq p_\tau \vee \neg p_\tau$, false $\triangleq \neg$true, $\psi_1 \to \psi_2 \triangleq \neg \psi_1 \vee \psi_2$, $\diamondsuit \psi \triangleq$ true$\mathcal{U} \psi$, and $\square \psi \triangleq \neg \diamondsuit \neg \psi$, where $\diamondsuit$ and $\square$ denote the '*eventually*' and '*always*', respectively. In $\varphi$, $\exists \tau$ means "along some trace $\tau$", and $\forall \tau$ means "along all traces $\tau$". We write $Vars(\varphi)$ for the set of trace variables appearing in $\varphi$. A formula is *closed* if all $\tau \in Vars(\varphi)$ are quantified exactly once.

Since our setting is episodic and each episode terminates at a finite time, we adopt the finite-trace semantics of HyperLTL (Brett et al., 2017). I.e., formulas are interpreted over finite trace assignments, given by a partial mapping $\Pi : Vars(\varphi) \rightharpoonup (2^{\mathsf{AP}})^*$ that assigns each $\tau \in Vars(\varphi)$ to a finite trace. Given a trace assignment $\Pi$, a trace variable $\tau$, and a finite trace $t \in (2^{\mathsf{AP}})^*$, we write $\Pi[\tau \mapsto t]$ for the assignment that agrees with $\Pi$ except that $\tau$ is mapped to $t$. We denote by $\Pi_\emptyset$ the empty trace assignment, and $t \in \Pi$ to refer to traces in the image of $\Pi$.

Given a closed HyperLTL formula $\varphi \triangleq \mathbb{Q}_1 \tau_1 \ldots \mathbb{Q}_n \tau_n. \psi$, where each $\mathbb{Q}_i \in \{\forall, \exists\}$, and $\psi$ is an inner LTL formula, an *interpretation* is a tuple $\mathcal{T} \triangleq \langle T_{\tau_i} \rangle_{i \in \{1, \ldots, |Vars(\varphi)|\}}$, where $T_{\tau_i} = \mathsf{Traces}(\mathcal{Z}_i)$ is a set of traces that may be assigned to the corresponding trace variable $\tau_i$. The satisfaction relation $\models$ maps a formula $\varphi$ to a model $(\mathcal{T}, \Pi, i)$, where $i \in \mathbb{Z}_{\geq 0}$ indicates the current evaluation position. Formally:

$$
\begin{aligned}
(\mathcal{T}, \Pi, 0) &\models \exists \tau. \psi & \text{iff} \quad & \exists t \in T_\tau. (\mathcal{T}, \Pi[\tau \to t], 0) \models \psi \\
(\mathcal{T}, \Pi, 0) &\models \forall \tau. \psi & \text{iff} \quad & \forall t \in T_\tau. (\mathcal{T}, \Pi[\tau \to t], 0) \models \psi \\
(\mathcal{T}, \Pi, i) &\models p_\tau & \text{iff} \quad & p \in \Pi(\tau)(i) \\
(\mathcal{T}, \Pi, i) &\models \neg \psi & \text{iff} \quad & (\mathcal{T}, \Pi, i) \not\models \psi \\
(\mathcal{T}, \Pi, i) &\models \psi_1 \vee \psi_2 & \text{iff} \quad & (\mathcal{T}, \Pi, i) \models \psi_1 \text{or} (\mathcal{T}, \Pi, i) \models \psi_2 \\
(\mathcal{T}, \Pi, i) &\models \bigcirc \psi & \text{iff} \quad & (\mathcal{T}, \Pi, i+1) \models \psi \\
& & & \text{and } \forall t \in \Pi. \, |t| > i + 1 \\
(\mathcal{T}, \Pi, i) &\models \psi_1 \, \mathcal{U} \, \psi_2 & \text{iff} \quad & \exists j \geq i \text{ with } j < \min_{t \in \Pi} |t|, \\
& & & \text{s.t. } (\mathcal{T}, \Pi, j) \models \psi_2 \\
& & & \text{and } \forall k \in [i, j), (\mathcal{T}, \Pi, k) \models \psi_1
\end{aligned}
$$

We say that an interpretation $\mathcal{T}$ satisfies a HyperLTL formula $\varphi$, written as $\mathcal{T} \models \varphi$, if $(\mathcal{T}, \Pi_\emptyset, 0) \models \varphi$.

**Example.** Consider the following arbitrary HyperLTL formula for the WildFire scenario:

$$\varphi_{\exp} \triangleq \forall \tau_1 \exists \tau_2. \, \mathsf{Dist}(\texttt{pos}_{\tau_1}, \texttt{pos}_{\tau_2}) < 3 \;\; \mathcal{U} \;\; (\texttt{pos}_{\tau_1} = i)$$

Here, FF corresponds to $\tau_1$ and Med corresponds to $\tau_2$. The predicate $\mathsf{Dist}(\texttt{pos}_{\tau_1}, \texttt{pos}_{\tau_2}) < 3$ enforces that they remain within distance 3, (until FF reaches cell $i$). Importantly, $\forall \exists$ captures inter-agent dependency by letting Med's decisions depend on FF's, whereas most specification-RL work focuses on $\forall \forall$, which cannot express this dependency.

## 4. Problem Statement

To bridge POMDP histories to HyperLTL interpretation, we define a function $\Phi : \mathcal{H}^* \times \mathcal{Z}^* \to [0, 1]$, which maps a action-observation history and a path to a scalar value in $[0, 1]$. We say that $\Phi$ is *path-consistent*, meaning that for any history $h$ generated by a latent path $\zeta$, $\zeta \in \arg\max_{\zeta' \in \mathcal{Z}^*} \Phi(h, \zeta')$. For a HyperLTL formula $\varphi \triangleq \mathbb{Q}_1 \tau_1 \ldots \mathbb{Q}_n \tau_n. \psi$, where each $\mathbb{Q}_i \in \{\forall, \exists\}$, let $\mathcal{D}_{\pi_i}$ denote the distribution over observation–action histories induced by a policy $\pi_i$ and let $\mathcal{H}_i \sim \mathcal{D}_{\pi_i}$ denote a *set* of histories sampled from this distribution, for $1 \leq i \leq n$ and $\mathcal{H}_i \subseteq \mathcal{H}^*$. The policy $\pi_i$ is associated with trace variable $\tau_i$, for each $i \in \{1, \ldots, |Vars(\varphi)|\}$, so that each trace variable ranges over the possible behaviors generated by its corresponding policy. The mapping $\Phi$ can be constructed by any technique such as belief-states constructions (Poupart & Boutilier, 2013; Rodriguez et al., 1999). For each trace variable $\tau_i \in Vars(\varphi)$, we introduce the set $\mathcal{Z}_i$ of paths,

---

**Problem Statement**

Given a POMDP $\mathcal{M}$ and a HyperLTL formula $\varphi = \mathbb{Q}_1\tau_1\ldots\mathbb{Q}_n\tau_n.\ \psi$, our goal is to identify a tuple of $n$ policies $\langle\pi_1^\star,\ldots,\pi_n^\star\rangle$, such that:

$$\langle\pi_i^\star\rangle_{i\in\{1,\ldots,n\}} \in \left[\underset{\langle\pi_i\rangle}{\arg\max}\ \mathbb{P}\Big[\langle\mathsf{Traces}\big(\bigcup_{h\in(\mathcal{H}_i\sim\mathcal{D}_{\pi_i})}\{\underset{\zeta\in\mathcal{Z}^*}{\arg\max}\ \Phi(h,\zeta)\}\big)\rangle \models \varphi\Big]\right]_{i\in\{1,\ldots,n\}}$$

where $\mathcal{D}_{\pi_1},\ldots,\mathcal{D}_{\pi_n}$ are the distributions over a set of histories drawn by policies $\pi_1,\ldots,\pi_n$, and $\bigcup_{h\in(\mathcal{H}_i\sim\mathcal{D}_{\pi_i})}\{\arg\max_{\zeta\in\mathcal{Z}^*}\Phi(h,\zeta)\}$ is a set of paths $\mathcal{Z}_i$ associated with policy $\pi_i$.

**Figure 3.** Formal problem statement of HᴙPOLE.

where each path $\zeta$ in $\mathcal{Z}_i$ has the highest probability for each $h \in \mathcal{H}_i$ drawn from distribution $\mathcal{D}_{\pi_i}$ using policy $\pi_i$:

$$\mathcal{Z}_i \triangleq \bigcup_{h\in(\mathcal{H}_i\sim\mathcal{D}_{\pi_i})}\big\{\underset{\zeta\in\mathcal{Z}^*}{\arg\max}\ \Phi(h,\zeta)\big\}$$

Then, the *family* of paths for each set $\mathcal{Z}_i$ of paths is defined as: $\mathbb{S} \triangleq \langle\mathcal{Z}_i\rangle_{i\in\{1,\ldots,|Vars(\varphi)|\}}$. We write, $\mathcal{T} = \mathsf{Traces}(\mathbb{S})$ as the tuple of sets of sampled traces. Also, we say a family of paths $\mathbb{S}$ (induced by each $\pi_\tau$ associated with each $\tau \in Vars(\varphi)$), satisfies a formula $\varphi$ if $\langle\mathsf{Traces}(\mathcal{Z}_i)\rangle_{i\in\{1,\ldots,|Vars(\varphi)|\}} \models \varphi$. The formal problem statement is shown in Fig. 3, where '$\star$' denotes optimality (e.g., $\pi^\star$ denotes an optimal policy). That is, the tuple of policies $\langle\pi_1^\star,\ldots,\pi_n^\star\rangle$ maximizes the probability $\mathbb{P}$ such that the generated tuple of sets of traces from $\mathcal{M}$ satisfies $\varphi$.

**Example.** Consider the POMDP in Fig. 2 and $\varphi_{\exp}$. Suppose FF samples histories with policy $\pi_1$, $\mathcal{H}_{\tau_1} = \{h_1^1, h_1^2\}$:

$$h_1^1 = \langle a\rangle \xrightarrow{\mathsf{R}} \langle b\rangle \xrightarrow{\mathsf{R}} \langle c,c\rangle \xrightarrow{\mathsf{U}} \langle f,f,f\rangle \xrightarrow{\mathsf{L}} \langle i,i\rangle$$
$$h_1^2 = \langle a\rangle \xrightarrow{\mathsf{R}} \langle b\rangle \xrightarrow{\mathsf{R}} \langle e\rangle \xrightarrow{\mathsf{L}} \langle h\rangle \xrightarrow{\mathsf{U}} \langle i,i\rangle$$

Likewise, agent Med, corresponding to $\tau_2$, samples histories using policy $\pi_2$, $\mathcal{H}_{\tau_2} = \{h_2^1, h_2^2\}$:

$$h_2^1 = \langle a\rangle \xrightarrow{\mathsf{U}} \langle d\rangle \xrightarrow{\mathsf{U}} \langle g,g\rangle \xrightarrow{\mathsf{R}} \langle d\rangle \xrightarrow{\mathsf{R}} \langle d\rangle$$
$$h_2^2 = \langle a\rangle \xrightarrow{\mathsf{U}} \langle b\rangle \xrightarrow{\mathsf{L}} \langle e\rangle \xrightarrow{\mathsf{R}} \langle f,f,f\rangle \xrightarrow{\mathsf{L}} \langle i,i\rangle$$

The set of paths associated with $\mathcal{H}_{\tau_i}$ is $\mathcal{Z}_{\tau_i} = \{\arg\max_{\zeta\in\mathcal{Z}^*}\Phi(h_i^1,\zeta), \arg\max_{\zeta\in\mathcal{Z}^*}\Phi(h_i^2,\zeta)\}$, for $i \in \{1,2\}$. We now compute the probability of satisfying $\varphi_{\exp}$ using $\mathcal{Z}_{\tau_1}$ and $\mathcal{Z}_{\tau_2}$ as follows, for $j \in \{1,2\}$:

$$\mathsf{Traces}(\langle\{\underset{\zeta\in\mathcal{Z}^*}{\arg\max}\ \Phi(h_1^j,\zeta)\},\mathcal{Z}_{\tau_2}\rangle) \models \varphi_{\exp}$$

Here, $\arg\max_{\zeta\in\mathcal{Z}^*}\Phi(h_2^2,\zeta)$ serves as a witness for $\tau_2$ in both satisfaction relations; hence the satisfaction probability of $\varphi_{\exp}$ under $\mathcal{H}_{\tau_1}$ and $\mathcal{H}_{\tau_2}$ is 1. However, if we replace the quantifiers in $\varphi_{\exp}$ with $\forall\forall$, the satisfaction probability drops to 0.75 (check App. B for details).

# 5. Algorithmic Details of HᴙPOLE

Our solution to the problem in Fig. 3 has three steps. First, we apply Skolemization (Skolem, 1879) to eliminate quanti-

fier alternation (Sec. 5.1). Next, we define robustness functions to quantify and optimize satisfaction of the HyperLTL formula (Sec. 5.2). Finally, we lift the shared-environment POMDP to a Decentralized POMDP and apply CTDE algorithms to learn policies to solve the problem in Fig. 3.

## 5.1. Step 1: HyperLTL Skolemization

Let $\varphi = \mathbb{Q}_1\tau_1\ldots\mathbb{Q}_n\tau_n.\ \psi(\tau_1,\ldots,\tau_n)$ be a HyperLTL formula. Following (Hsu et al., 2025), we Skolemize $\varphi$ to eliminate quantifier alternation. Let $\mathbb{Q}^\exists = \{i \mid \mathbb{Q}_i = \exists\}$ and $\mathbb{Q}^\forall = \{j \mid \mathbb{Q}_j = \forall\}$. For each $i \in \mathbb{Q}^\exists$, let $\mathbb{Q}_i^\forall = \{j \in \mathbb{Q}^\forall \mid j < i\}$ denote the set of indices of all *preceding* universal quantifiers. For each $i \in \mathbb{Q}^\exists$, a Skolem function $\mathbf{f}_i : \mathcal{T}^{|\mathbb{Q}_i^\forall|} \to \mathcal{T}$ is defined, reducing to a constant function when $\mathbb{Q}_i^\forall = \emptyset$. A trace assignment $\Pi$ is consistent with $\mathbf{f}_i$ if, for all $i \in \mathbb{Q}^\exists$, $\Pi(\tau_j) \in \mathcal{T}$ for all $j \in \mathbb{Q}_i^\forall$ and $\Pi(\tau_i) = \mathbf{f}_i\big(\Pi(\tau_{i_1}),\Pi(\tau_{i_2}),\ldots,\Pi(\tau_{i_{|\mathbb{Q}_i^\forall|}})\big)$ for all $i \in \mathbb{Q}^\exists$, where $\mathbb{Q}_i^\forall = \{i_1 < \cdots < i_{|\mathbb{Q}_i^\forall|}\}$. If $(\mathcal{T},\Pi,0) \models \varphi$ holds for all trace assignments consistent with all $\mathbf{f}_i$, then each $\mathbf{f}_i$ witnesses the satisfaction of $\varphi$ (Winter & Zimmermann, 2025). For the inner LTL formula $\psi$, we obtain $\mathbf{Skolem}(\psi)$ by substituting each proposition $p_{\tau_i}$ with $p_{\mathbf{f}_i}$ for all $p \in \mathsf{AP}$ and $i \in \mathbb{Q}^\exists$, thereby instantiating existential traces with their Skolem witnesses.x The Skolemized formula is:

$$\mathbf{Skolem}(\varphi) = \underbrace{\exists\mathbf{f}_i(\tau_{i_1},\ldots,\tau_{i_{|\mathbb{Q}_i^\forall|}}).}_{\text{for each } i\in\mathbb{Q}^\exists}\ \underbrace{\forall\tau_j.}_{\text{for each } j\in\mathbb{Q}^\forall}\ \mathbf{Skolem}(\psi)$$

Based on this transformation, we rewrite the problem statement in Fig. 3 of Sec. 4. The updated statement is shown in Eq. (1), where $Img(\mathbf{f}_i)$ denotes the set of traces obtained by applying the Skolem function to the preceding universally quantified traces (as detailed in App. C.1).

## 5.2. Step 2: Learning with Quantitative Semantics

We reformulate the HyperLTL satisfaction problem as an optimization problem under its quantitative semantics, where the Skolemized formula $\mathbf{Skolem}(\varphi)$ is evaluated on tuples of histories $\langle h_1,\ldots,h_n\rangle$ sampled from the POMDP $\mathcal{M}$. Let $\mathbb{R}$ be the set of real numbers, $\Psi$ the set of all LTL formulas, and $f : 2^{\mathsf{AP}} \to \mathbb{R}$ be a *valuation function* assigning real values to sets of atomic propositions. For a state $s \in S$ of $\mathcal{M}$, quantitative semantics are defined by predicates of the form $f\big(L(s)\big) < c$, where $c$ is a user-specified threshold (see Fig. 9 in Appendix for full semantics). The robustness function $\rho : \mathsf{Traces}(\mathcal{Z}^*) \times \Psi \to \mathbb{R}$ assigns a real-valued

$$\langle \pi_j^\star \rangle_j \in \left[ \arg \max_{\langle \pi_j \rangle} \mathbb{P}\left[ \langle Img(\mathbf{f}_i) \rangle \bowtie \langle \mathsf{Traces}( \bigcup_{h \in (\mathcal{H}_{ij} \sim \mathcal{D}_{\pi_{ij}})} \{ \arg \max_{\zeta \in \mathcal{Z}^*} \Phi(h, \zeta) \}) \rangle \models \mathbf{Skolem}(\varphi) \right] \right]_{i \in \mathbb{Q}^\exists, j \in \mathbb{Q}^\forall} \quad (1)$$

$$\langle \pi_\ell^\star \rangle_\ell \in \left[ \arg \max_{\langle \pi_\ell \rangle} \mathbb{P}\left[ \rho\left( \mathsf{zip}\langle \mathsf{Tr}((\arg \max_{\zeta \in \mathcal{Z}^*} \Phi(h_\ell \sim \mathcal{D}_{\pi_\ell}, \zeta))_{[0:k_l]}) \rangle), \psi \right) \xrightarrow{\star} \rho_{max} \right] \right]_{\ell \in \{1, \ldots n\}} \quad (2)$$

$$\langle \pi_i^\star \rangle \sqcup \langle \pi_j^\star \rangle \in \left[ \arg \max_{\langle \pi_i \rangle \sqcup \langle \pi_j \rangle} \mathbb{P}\left[ \rho\left( \mathsf{zip}(\langle \mathsf{Tr}((\arg \max_{\zeta \in \mathcal{Z}^*} \Phi(h_i \sim \mathcal{D}_{\pi_i}, \zeta))_{[0:k_i]}) \rangle \sqcup \right. \right. \right.$$
$$\left. \left. \left. \langle \mathsf{Tr}((\arg \max_{\zeta \in \mathcal{Z}^*} \Phi(h_j \sim \mathcal{D}_{\pi_j}, \zeta))_{[0:k_j]}) \rangle), \mathbf{Skolem}(\psi) \right) \xrightarrow{\star} \rho_{max} \right] \right]_{i \in \mathbb{Q}^\exists, j \in \mathbb{Q}^\forall} \quad (3)$$

score to a finite trace and an LTL formula, bounded by $\rho_{max}$ and $\rho_{min}$. Formally, given an LTL formula $\psi$ and a POMDP $\mathcal{M}$, we formulate the policy synthesis problem as:

$$\pi^\star \in \arg \max_\pi \mathbb{P}_{h \sim \mathcal{D}_\pi} \left[ \rho\left( \mathsf{Tr}((\arg \max_{\zeta \in \mathcal{Z}^*} \Phi(h, \zeta))_{[0:k]}), \psi \right) \xrightarrow{\star} \rho_{max} \right]$$

where $\xrightarrow{\star}$ means convergence. I.e., $\pi^\star$ maximizes the probability that paths induced by histories under $\pi$ satisfy $\psi$. Next, to compute robustness over multiple (universally quantified) traces, we define a zip operator as follows:

$$\mathsf{zip}(\langle t_\ell \rangle_{\ell \in \{1, \ldots, n\}}) \triangleq \langle t_\ell(0) \rangle_{\ell \in \{1, \ldots, n\}} \cdots \langle t_\ell(k) \rangle_{\ell \in \{1, \ldots, n\}}$$

where $k \triangleq \min_{\ell \in \{1, \ldots, n\}} m_\ell$. E.g., given $t_1 = a_1 a_2 a_3$ and $t_2 = b_1 b_2 b_3$, their zipped trace is $\mathsf{zip}(\langle t_1, t_2 \rangle) = \langle a_1, b_1 \rangle \langle a_2, b_2 \rangle \langle a_3, b_3 \rangle$. Thus, the optimization problem of computing a tuple of policies $\langle \pi_1^\star, \ldots, \pi_n^\star \rangle$ that maximizes robustness can be stated as Eq. (2).

Now, let $\langle t_i \rangle_{i \in I}$ and $\langle t_j \rangle_{j \in J}$ be two trace tuples, where $I \cup J = \{1, \ldots, n\}$ and $I \cap J = \emptyset$. To preserve index order when combining them, we use $\sqcup$, which produces a single tuple ordered by path indices. Based on this construction, satisfaction of the inner LTL body $\psi$ is defined over tuples of histories. For each $i \in \mathbb{Q}^\exists$ and $j \in \mathbb{Q}^\forall$, a tuple $\langle h_1, \ldots, h_n \rangle$ satisfies $\psi$ if and only if, after combining the paths associated with histories via ordered union and zipping them into a joint trace, the resulting robustness score converges to $\rho_{max}$ (details in App. C.2). Accordingly, we formulate the optimization problem in (3).

Observe Eq. (3), the robustness value $\rho$ for an $\exists$-quantified trace instantiated by a Skolem function depends on whether the trace induced by $h_i \sim \mathcal{D}_{\pi_i}$ can serve as a valid witness for the preceding $\forall$-quantified traces. Concretely, the robustness of a Skolem witness $\mathbf{f}_i$ with respect to $\psi$ is defined as $\rho(\mathbf{f}_i, \psi) \triangleq \rho(\mathbf{f}_i(\mathsf{Tr}(\zeta_{i_1}), \ldots, \mathsf{Tr}(\zeta_{i_{|\mathbb{Q}_i^\forall|}})), \psi)$. As a result, the optimization of $\langle \pi_i^\star \rangle$ is inherently coupled with $\langle \pi_j^\star \rangle$, which is crucial for correctly capturing the semantics of HyperLTL formulas with quantifier alternation.

### Theorem 5.1

Given a POMDP $\mathcal{M}$ and a HyperLTL formula $\varphi$, if $\Phi$ is path-consistent, then any tuple of policies $\langle \pi_i^\star \rangle_{i \in \mathbb{Q}^\exists} \sqcup \langle \pi_j^\star \rangle_{j \in \mathbb{Q}^\forall}$ that optimizes the Skolemized formula $\mathbf{Skolem}(\varphi)$ also optimizes the probability of satisfying $\varphi$ in $\mathcal{M}$ as defined in Fig. 3.

### 5.3. Step 3: MARL for HyperLTL

Now, we aim to synthesize policies for the universally quantified path variables and to learn Skolem functions that witness the existential quantifiers in the Skolemized Hyper-LTL formula. To this end, we adopt value-based CTDE paradigm, i.e., VDN (Sunehag et al., 2017), QMIX (Rashid et al., 2018), and QTRAN (Son et al., 2019). In CTDE, training leverages global information, such as other agents' observations or the global state, while execution is decentralized and each agent acts based on its local observation.

Our objective is to compute tuples of optimal policies $\langle \pi_i^\star \rangle_{i \in \mathbb{Q}^\exists}$ and $\langle \pi_j^\star \rangle_{j \in \mathbb{Q}^\forall}$ that solve (3). For each $j \in \mathbb{Q}^\forall$, we construct an optimal policy $\pi_j^\star(h_{j[0:k]})$ that depends solely on $h_j$. For each $i \in \mathbb{Q}^\exists$, the corresponding policy takes a history induced by the associated Skolem witness as an input. To formalize this, we introduce a trace-to-history consistency map $\mathfrak{H} : \mathcal{T} \to 2^\mathcal{H}$, where $\mathfrak{H}(t) \triangleq \{ h \in \mathcal{H} \mid \mathsf{Tr}(\arg \max_{\zeta \in \mathcal{Z}^*} \Phi(h, \zeta)) = t \}$. That is, $\mathfrak{H}(t)$ denotes the set of histories whose most likely latent path under $\Phi$ induces trace $t$. We assume that $\mathfrak{H}(t) \neq \emptyset$ for every $t \in \mathcal{T}$ under consideration. Since policies require a single history as input, we fix a function Sel, that maps a nonempty set $X$ to an arbitrary member of $X$, and introduce $\Omega : \mathcal{T} \to \mathcal{H}$, and defined as, $\Omega(t) \triangleq \mathsf{Sel}(\mathfrak{H}(t))$. Finally, for each $i \in \mathbb{Q}^\exists$, we construct an optimal policy that takes a history induced by the corresponding Skolem witness as an input, namely, $\pi_i^\star(\Omega(\mathbf{f}_i(\mathsf{Tr}(\zeta_{i_1[0:k]}), \ldots, \mathsf{Tr}(\zeta_{i_{|\mathbb{Q}_i^\forall|}[0:k]}))))$. This construction implies that the decisions of the optimal policies associated with existential quantifiers depend on the optimal policies of the preceding universal quantifiers for capturing the inter-agent dependencies.

**From POMDP to Dec-POMDP.** Value-based CTDE algorithms are defined over Decentralized POMDPs (Dec-POMDP) (Oliehoek et al., 2016), represented as a tuple $G = \langle \mathcal{S}, \mathcal{I}, A, \mathcal{P}, \mathcal{R}, \mathcal{O}, \mathfrak{O}, n, \mathsf{AP}, L, \gamma \rangle$. It is important to note that a POMDP is a special case of a Dec-POMDP (Lauri et al., 2020), and our framework simultaneously samples all the histories associated with each quantifier from the POMDP. To make the shared POMDP compatible with CTDE algorithms, we formally lift the shared POMDP $\mathcal{M}$ for $n$ agents into a Dec-POMDP. In this transformation, the state space remains the same, i.e., $\mathcal{S} \triangleq S$ and $\mathcal{I} \in \Delta(\mathcal{S})$. The joint action space is defined as

the Cartesian product of the individual agent action spaces, i.e., $\mathcal{A} \triangleq A^n$ (we assume the same action space for all agents), and a joint action is given by $\mathbf{a} \triangleq \langle a_\ell \rangle_{\ell \in \{1,\dots,n\}}$. The transition function $\mathcal{P}(s' \mid s, \mathbf{a})$ specifies the conditional probability of transitioning to state $s' \in \mathcal{S}$ after taking joint action $\mathbf{a}$ in $s \in \mathcal{S}$. When taking $\mathbf{a}$ in $s$, the agents receive a scalar reward $\mathcal{R}(s_k, \mathbf{a}_k)$. The components $\gamma$, AP, and $L$ are defined as in the POMDP case. The observation space in $G$ is defined as $\mathfrak{O} \triangleq O^n$, where each agent $\ell$ receives an individual observation from $o_\ell \in O$ based on the observation probability function $\mathcal{O}(o_\ell \mid s, a_\ell)$. The joint history is denoted by $\mathbf{h} \triangleq \langle h_\ell \rangle_{\ell \in \{1,\dots n\}}$, where $h_\ell$ is the local history of agent $\ell \in \{1, \dots, n\}$. Each $h_\ell$ is obtained using $\mathcal{O}$.

**Robustness Values as Reward Signals.** Let the state of a zipped path at position $k$ as $s_k = \mathsf{zip}_k(\langle t_\ell \rangle_{\ell \in \{1,\dots,n\}})$. The immediate reward signal at step $k$ is based on the scalar robustness value of the zipped trace from Eq. (3) defined as:

$$\mathcal{R}(s_k, \mathbf{a}_k) \triangleq \Big[ \rho\Big( \mathsf{zip}\big( \langle \mathsf{Tr}((\arg\max_{\zeta \in \mathcal{Z}^*} \Phi(h_i, \zeta))_{[0:k+1]}) \rangle \sqcup$$

$$\langle \mathsf{Tr}((\arg\max_{\zeta \in \mathcal{Z}^*} \Phi(h_j, \zeta))_{[0:k+1]}) \rangle \big), \mathbf{Skolem}(\psi) \Big) \Big]_{i \in \mathbb{Q}^\exists, j \in \mathbb{Q}^\forall}$$

Since paths are defined over state–action sequences, $\mathbf{a}_k$ is included in the zipped paths prefix to evaluate $\rho$. Moreover, because Dec-POMDP rewards are defined over the underlying state and joint action, we are allowed to assume that $\Phi$ is *path-consistent* to compute robustness-based reward and feedback from observation histories. This assumption is used only for reward computation during training, and is not needed to generate the optimal policies.

**MARL Optimization Problem.** The formal optimization problem is to find a joint policy $\boldsymbol{\pi}(\mathbf{h}) \triangleq \langle \pi_\ell(h_\ell) \rangle_{\ell \in \{1,\dots,n\}}$ that maximizes the joint value function, defined as, $\mathcal{V}^{\boldsymbol{\pi}}(\mathbf{h}_{[0:k]}) = \mathbb{E}[\mathcal{R}(s_k, \mathbf{a}_k) + \gamma \mathcal{V}^{\boldsymbol{\pi}}(\mathbf{h}_{[0:k+1]})]$, where $\mathbf{a}_k = \boldsymbol{\pi}(\mathbf{h}_{[0:k]})$.

As an alternative, many RL methods work with the joint history–action value function (Q-function) $\mathbf{Q}^{\boldsymbol{\pi}}(\mathbf{h}, \mathbf{a})$, which evaluates taking joint action $\mathbf{a}$ at joint history $\mathbf{h}$ and continuing according to $\boldsymbol{\pi}$, $\mathbf{Q}^{\boldsymbol{\pi}}(\mathbf{h}_{[0:k]}, \mathbf{a}_k) = \mathbb{E}[\mathcal{R}(s_k, \mathbf{a}_k) + \gamma \mathbb{E}[\mathbf{Q}^{\boldsymbol{\pi}}(\mathbf{h}_{[0:k+1]}, \boldsymbol{\pi}(\mathbf{h}_{[0:k+1]}))] \mid \mathbf{a}_k]$.

CTDE methods learn per-agent utilities that are combined (e.g., via a mixing network) into a joint Q-function $\mathbf{Q}$ during centralized training. At execution time, agents act greedily using only local utilities, enabling decentralized control. Formally, CTDE aims to approximate $\mathbf{Q}^*(\mathbf{h}, \mathbf{a}) = \max_{\boldsymbol{\pi}} \mathbf{Q}^{\boldsymbol{\pi}}(\mathbf{h}, \mathbf{a})$ and induce the joint optimal policy $\boldsymbol{\pi}^\star(\mathbf{h}) \in \arg\max_{\mathbf{a}} \mathbf{Q}^*(\mathbf{h}, \mathbf{a})$.

**Constructing Policies Using CTDE.** From the given joint policy $\boldsymbol{\pi} \triangleq \langle \pi_\ell \rangle_{\ell \in \{1,\dots,n\}}$, we construct two tuples of policies to the corresponding indices: $\langle \pi_i \rangle_{i \in \mathbb{Q}^\exists}$ and $\langle \pi_j \rangle_{j \in \mathbb{Q}^\forall}$.

In CTDE algorithms, each agent (corresponding to a trace variable in our setting) selects actions during execution

based solely on its own local observation history. Thus, for the policies associated with universally quantified traces, denoted by $\langle \pi_j \rangle_{j \in \mathbb{Q}^\forall}$, where each policy takes as input its local history $h_{j[0:k]}$, we can directly apply CTDE methods. In particular, we use robustness values as reward signals to approximate $\mathbf{Q}$ during centralized training, enabling the iterative construction of the policies $\langle \pi_j \rangle_{j \in \mathbb{Q}^\forall}$.

For policies associated with existentially quantified traces, denoted by $\langle \pi_i \rangle_{i \in \mathbb{Q}^\exists}$, each agent's policy requires as input the history induced by the corresponding Skolem witness, which is produced by the Skolem function $\mathbf{f}_i$. To compute $\mathbf{f}_i$, access to the paths associated with the preceding universally quantified trace variables, namely $\zeta_{i_1[0:k]}, \dots, \zeta_{i_{|\mathbb{Q}_i^\forall|}[0:k]}$, is required. Such information is available during the centralized training, but not during decentralized execution; hence, during training, we construct a replay buffer $\mathcal{D}$ that stores sequences of the form $(o_i, s_{i_1}, \dots, s_{i_{|\mathbb{Q}_i^\forall|}})$ for each $i \in \mathbb{Q}^\exists$.

Next, we train $f_{\mathrm{enc\text{-}dec}} : \mathcal{H} \to \mathcal{Z}^{|\mathbb{Q}_i^\forall|}$ model. During centralized training, we sample sequences from the replay buffer $\mathcal{D}$ and use them to train $f_{\mathrm{enc\text{-}dec}}$ to predict the paths $(\hat{\zeta}_{i_1[0:k]}, \dots, \hat{\zeta}_{i_{|\mathbb{Q}_i^\forall|}[0:k]})$ given $h_{i[0:k]}$. The predicted paths serve as inputs to the Skolem function $\mathbf{f}_i$, enabling the construction of policies for existentially quantified traces; in particular, for each $i \in \mathbb{Q}^\exists$, we have, $\pi_i(\Omega\,(\mathbf{f}_i(\mathsf{Tr}(\hat{\zeta}_{i_1[0:k]}), \dots, \mathsf{Tr}(\hat{\zeta}_{i_{|\mathbb{Q}_i^\forall|}[0:k]}))))$.

Up to now, we have described how to learn $\langle \pi_i \rangle_{i \in \mathbb{Q}^\exists}$ and $\langle \pi_j \rangle_{j \in \mathbb{Q}^\forall}$ using CTDE. However, it is important that CTDE algorithms are designed to approximate a joint action–value function and to induce an optimal joint policy $\boldsymbol{\pi}^\star$. This objective does not, in general, imply that the resulting $\langle \pi_i^\star \rangle_{i \in \mathbb{Q}^\exists} \sqcup \langle \pi_j^\star \rangle_{j \in \mathbb{Q}^\forall}$ is optimal in all CTDE methods. In (Son et al., 2019), a formal property called *Individual–Global–Max* (IGM) is introduced. IGM property states that maximizing the joint Q-function is equivalent to independently maximizing each agent's local function.

**Definition 5.2.** For a joint action–value function $\mathbf{Q}(\mathbf{h}, \mathbf{a})$, if there exist a tuple of functions $\langle Q_\ell \rangle_{\ell \in \{1,\dots,n\}}$ such that:

$$\arg\max_{\mathbf{a}} \mathbf{Q}(\mathbf{h}, \mathbf{a}) = \langle \arg\max_{a_\ell} Q_\ell(h_\ell, a_\ell) \rangle_{\ell \in \{1,\dots,n\}}$$

where $\langle Q_\ell \rangle_{\ell \in \{1,\dots,n\}}$ satisfies *IGM* for $\mathbf{Q}$ given $\mathbf{h}$. Notice that, among the CTDE algorithms we consider, only QTRAN satisfies the IGM property.

---

**Theorem 5.3**

Let $\mathcal{M}$ be a POMDP and $\varphi$ a HyperLTL formula. If a CTDE algorithm learns an optimal joint action–value function $\mathbf{Q}^\star$ satisfying the IGM property, then $\langle \pi_i^\star \rangle_{i \in \mathbb{Q}^\exists}$ and $\langle \pi_j^\star \rangle_{j \in \mathbb{Q}^\forall}$ induced by the joint policy $\boldsymbol{\pi}^\star$ will optimize the satisfaction probability of $\mathbf{Skolem}(\varphi)$.

---

Theorem 5.3 establishes the premise of Theorem 5.1, which in turn solves the original problem Fig. 3 (proofs in App. A).

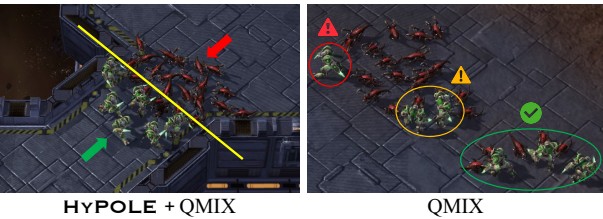

**Figure 4.** HYPOLE +QMIX vs. QMIX on SMAC corridor map.

## 6. Experiments and Results

**Implementation.** HYPOLE is implemented on top of PyMARL repository (Samvelyan et al., 2019). We start with a Skolemized HyperLTL formula. Using the quantitative semantics described in Sec. 5.2, we construct robustness functions and use the resulting robustness values as reward signals in a cooperative MARL environment. Our objective is to synthesize policies $\pi_\ell$ for each agent $\ell \in \{1, \ldots, n\}$, depending on the quantifier associated with its trace variable. For universally quantified traces, i.e., policies $\langle \pi_j \rangle_{j \in \mathbb{Q}^\forall}$, each policy receives only the agent's local history. In contrast, for existentially quantified traces, i.e., policies $\langle \pi_i \rangle_{i \in \mathbb{Q}^\exists}$, the policy input is the history induced by the output of the corresponding Skolem function $\mathbf{f}_i$.

We train an encoder-decoder model to predict the traces of the preceding universally quantified variables. During centralized training, mini-batches sampled from the replay buffer $\mathcal{D}$ are used to train the encoder-decoder model every $\beta$ steps. The encoder-decoder model follows a feedforward autoencoder-style architecture. We then employ value-based CTDE algorithms, namely VDN, QMIX, and QTRAN to construct the policies. In addition, we evaluate Independent Q-Learning (IQL) (Tan, 1993). For IQL, we consider only $\forall\forall$-form HyperLTL specifications, which evaluates using local observations alone. Additional details on the experimental setting are provided in App. D.

**Case Studies.** We conduct our experiments on scenarios from the StarCraft Multi-Agent Challenge (SMAC) (Samvelyan et al., 2019), which is based on the strategy game StarCraft II and focuses on cooperative micromanagement tasks. In addition, we use MessySMAC (Phan et al., 2023), which introduces observation stochasticity $\phi$ and randomized initialization with $K$ random steps before each episode starts. This case study adds significant challenges for finding optimal policies in SMAC. Finally, we extend the WildFire benchmark introduced in (Hsu et al., 2025) to a partially observable setting. More details on SMAC, MessySMAC, and WildFire are provided in Apps. E and F.

**Experimental Organization.** We evaluate HYPOLE on nine SMAC scenarios, including both hard and super-hard scenarios (see Table 1 in the Appendix), five MessySMAC scenarios, and four WildFire scenarios (see Table 2 in the Appendix). We (1) assess the effect of using robustness val-

ues from $\forall\forall$-HyperLTL specifications as reward signals in MARL, compared to baseline MARL algorithms using standard shaped rewards; (2) compare $\forall\exists$-HyperLTL specifications with $\forall\forall$-HyperLTL specifications in terms of learning efficiency and performance; (3) investigate how HyperLTL enables expressing distinct combat tactics, and (4) discuss HyperLTL specifications that fail to provide effective guidance for learning. All specifications used in our experiments are presented in App. G.

**HYPOLE vs. Shaped Rewards.** We compare the robustness values generated by HYPOLE using $\forall\forall$-HyperLTL specifications with the shaped reward functions introduced in (Samvelyan et al., 2019). Fig. 5 reports the median test win rate in SMAC (i.e., elimination of all enemy agents) of HYPOLE +MARL, where MARL $\in$ {VDN, QMIX, QTRAN, IQL}, compared against vanilla MARL using shaped rewards. In Fig. 5a, we present our results for the $\varphi_{\text{focus-fire}}$ on the 3s5z map, where $\varphi_{\text{focus-fire}}$ encourages allied agents to focus fire on enemies. Across all settings for this map, HYPOLE +MARL outperforms the vanilla MARL baselines. Notably, HYPOLE +QMIX (orange line) initially underperforms vanilla QMIX (blue line), but after $\approx 700$K environment steps the agents learn the focus fire tactic and subsequently surpass vanilla QMIX. Similarly, on the 8m map Fig. 5b, HYPOLE +MARL using $\varphi_{\text{focus-fire}}$ outperforms the vanilla MARL baselines in all cases. In particular, HYPOLE +IQL (gray line) not only surpasses vanilla IQL (pink line), but also outperforms vanilla QTRAN (purple line), despite QTRAN being a more complex CTDE method. Moreover, HYPOLE +IQL remains competitive with vanilla VDN (green line) and QMIX (blue line). We also evaluate $\varphi_{\text{focus-fire}}$ on bane_vs_bane map Fig. 5c, where HYPOLE +QMIX (orange line) and HYPOLE +VDN (red line) significantly outperform vanilla QMIX (blue line) and VDN (green line).

Fig. 5d evaluates HYPOLE +MARL using $\varphi_{\text{kite}}$ formula on 5m_vs_6m, where $\varphi_{\text{kite}}$ encourages agents to attack only when their weapon is ready. For all cases on this map, HYPOLE +MARL competes with the vanilla MARL baselines. In Fig. 5e, we evaluates the $\varphi_{\text{medivac}}$ on MMM map, where $\varphi_{\text{medivac}}$ encourages Medivac agents to move toward low-health allies to heal them. In all cases, HYPOLE +MARL outperforms the vanilla baselines. In particular, HYPOLE +QTRAN (brown line) and HYPOLE +IQL (gray line) substantially improve over vanilla QTRAN (purple line) and IQL (pink line), respectively. We find HYPOLE particularly promising on the corridor map (Fig. 5f), which is widely regarded as one of the hardest SMAC maps. We use $\varphi_{\text{corridor}}$ formula, which encourages agents to hold the choke point and damage enemies from that position. Using this formula, HYPOLE +QMIX (orange line) achieves a median win rate of up to $\approx 60\%$, substantially outperforming vanilla QMIX (blue line), which achieves a win rate of less than $5\%$. To

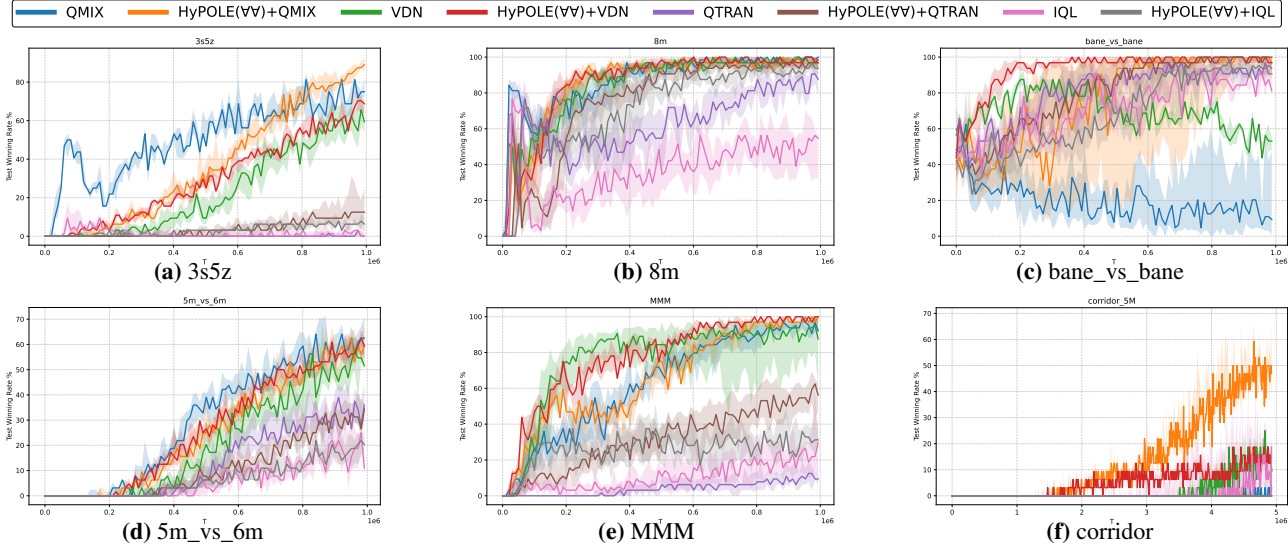

**Figure 5.** Learning curves on different SMAC maps, showing median test win rate with 25-75% percentile.

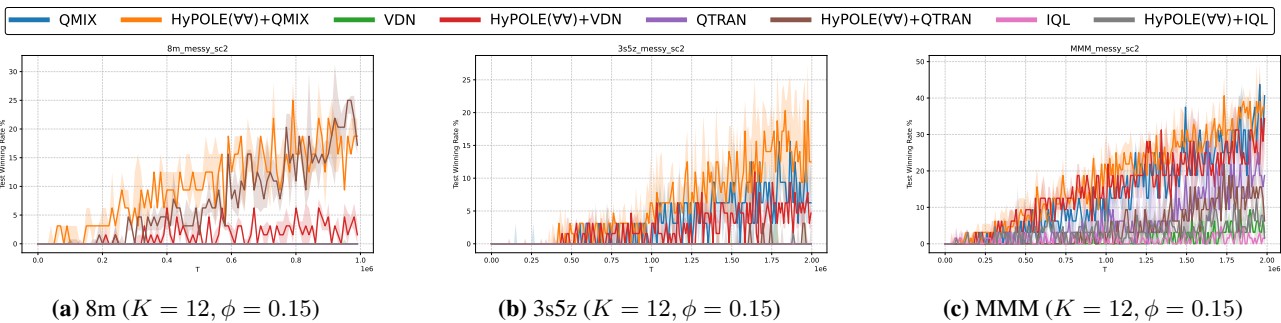

**Figure 6.** Learning curves on different MessySMAC maps, showing median test win rate with 25-75% percentile.

illustrate this, Fig. 4 shows snapshots of agent behavior after 5M training steps. Under HYPOLE +QMIX, agents form a defensive line at the choke point. In contrast, vanilla QMIX agents lose formation and split into three groups. One group of three agents, marked by ✅, is surrounded by five enemies and may still survive because Zealots are stronger than Zerglings. However, the single agent marked by ⚠️ is surrounded by more than ten Zerglings and is likely to be defeated. This then exposes the two agents marked by ⚠️, leading to the loss of three allies and eventually the remaining agents, who are heavily outnumbered.

We also compare HYPOLE +MARL against vanilla MARL on MessySMAC, with results shown in Fig. 6. In 8m map Fig. 6a, we use the $\varphi_{\text{focus-fire}}$ formula. Across all cases, HYPOLE +CTDE outperforms vanilla CTDE. In particular, HYPOLE +QTRAN (brown line) and HYPOLE +QMIX (orange line) achieve win rates of up to 25%, whereas the vanilla baselines fail to achieve any wins. Similarly, we use $\varphi_{\text{focus-fire}}$ in 3s5z map Fig. 6b. In this scenario, HYPOLE +VDN (red line) outperforms vanilla VDN, which fails to

achieve any wins. Moreover, HYPOLE +QMIX (orange line) performs slightly better than vanilla QMIX (blue line). In MMM map Fig. 6c, we use the $\varphi_{\text{medivac}}$ formula. Here, HYPOLE +VDN (red line) achieves a win rate of up to $\approx 35\%$, substantially outperforming vanilla VDN (green line), which remains below 10%. The remaining SMAC and MessySMAC maps, together with the full WildFire comparison against hand-built rewards, are provided in App. H.1.

$\forall\exists$ **vs.** $\forall\forall$ **Specifications.** We study the effect of using $\forall\exists$ variants of the $\varphi_{\text{medivac}}$ and $\varphi_{\text{wildfire}}$ specifications, and show that they can lead to improved policies by expanding the policy search space. In the MMM scenario in SMAC and MessySMAC, there is an inter-agent dependency between the Medivac and the remaining agents. To capture this, we change the quantifier associated with the Medivac from $\forall$ to $\exists$ in $\varphi_{\text{medivac}}$. In MMM scenario in SMAC Fig. 7a, HYPOLE $(\forall\exists)$+QMIX (orange line) and HYPOLE $(\forall\exists)$ +VDN (red line) outperform HYPOLE $(\forall\forall)$+QMIX (blue line) and HYPOLE $(\forall\forall)$+VDN (green line). For the MessySMAC benchmark, Fig. 6c shows that HYPOLE $(\forall\forall)$+QMIX has

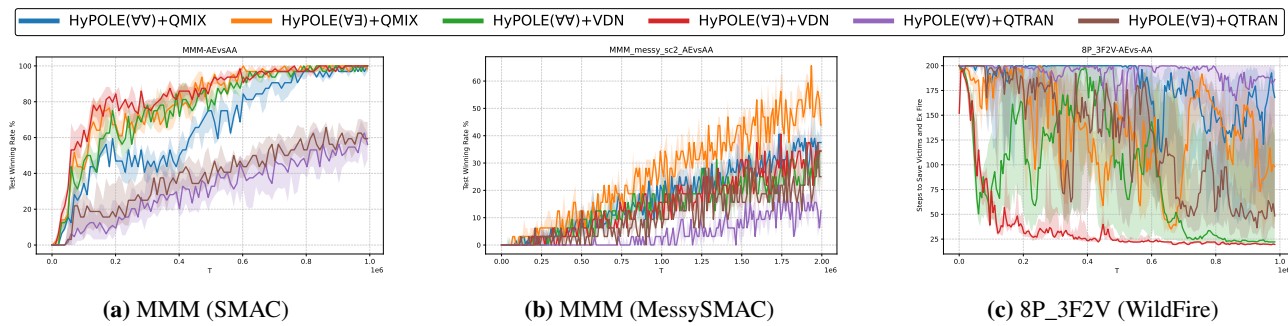

**(a)** MMM (SMAC)  **(b)** MMM (MessySMAC)  **(c)** 8P_3F2V (WildFire)

**Figure 7.** ∀∀ vs. ∀∃ learning curves across MessySMAC, SMAC, and WildFire, showing median test win rate with 25–75% percentiles.

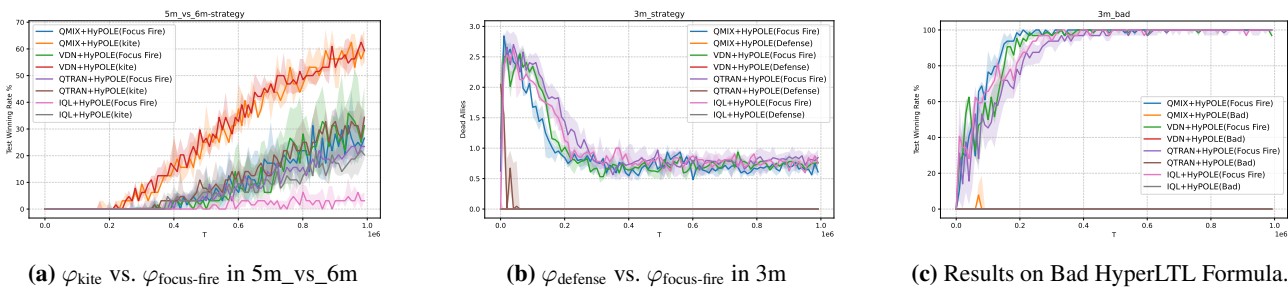

**(a)** $\varphi_{\text{kite}}$ vs. $\varphi_{\text{focus-fire}}$ in 5m_vs_6m  **(b)** $\varphi_{\text{defense}}$ vs. $\varphi_{\text{focus-fire}}$ in 3m  **(c)** Results on Bad HyperLTL Formula.

**Figure 8.** Expressing tactics using HYPOLE, and results on a bad HyperLTL formula.

only a slight edge over the vanilla baseline. In contrast, Fig. 7b shows that HYPOLE +QMIX(∀∃) (orange line) performs better than HYPOLE +QMIX(∀∀) (blue line) and, consequently, the vanilla QMIX baseline. Similarly, in Fig. 6c, HYPOLE +QTRAN(∀∀) performs worse than the vanilla baseline. However, using HYPOLE +QTRAN(∀∃) (brown line) in Fig. 7b improves the performance of HY-POLE. In Fig. 7c, we observe that the ∀∃ version of the $\varphi_{\text{wildfire}}$ formula helps HYPOLE achieve better results in the WildFire 8P_3F2V scenario. Specifically, HYPOLE +VDN(∀∃) (red line) performs substantially better than HY-POLE +VDN(∀∀) (green line), converging after 500K training steps compared to 800K steps. Additional results comparing ∀∃ and ∀∀ are provided in App. H.2.

**Tactics as HyperLTL Formulas.** A key advantage of HYPOLE is learning policies that maximize satisfaction of HyperLTL-encoded tactics. In Fig. 8a, we study the 5m_vs_6m scenario and show that the kiting ($\varphi_{\text{kite}}$) yields consistently higher win rates than the focus firing ($\varphi_{\text{focus-fire}}$), and this trend is consistently evident in all cases. In 3m map Fig. 8b, we further demonstrate the expressiveness of HYPOLE, by specifying a defensive tactic using $\varphi_{\text{defense}}$, which encourages allied agents to retreat from enemies and results in zero ally casualties. This experiment shows that diverse tactics can be encoded in HyperLTL and compiled into policies via HYPOLE.

**Quality of HyperLTL Specifications.** We also study the effect of weakening a HyperLTL specification by removing a subformula from the $\varphi_{\text{focus-fire}}$, resulting in the $\varphi_{\text{bad}}$. This modification eliminates the constraint that encourages

agents to shoot enemies. In Fig. 8c, HYPOLE with $\varphi_{\text{bad}}$ formula achieves an almost zero win rate. This shows that the performance of HYPOLE strongly depends on the quality and completeness of the underlying HyperLTL specification.

## 7. Conclusion

We presented HYPOLE, a specification-guided MARL framework that enables model-free synthesis of decentralized policies from HyperLTL specifications over POMDPs. HYPOLE effectively handles complex multi-agent objectives under partial observability by leveraging the semantics of HyperLTL to guide the learning process by off-the-shelf MARL algorithms. HYPOLE is fully implemented and we showed superior performance over baselines, namely, vanilla versions of VDN, QMIX, QTRAN, and IQL, across a set of diverse case studies.

**Limitations.** Admittedly, the main challenge in using HY-POLE is writing precise HyperLTL specifications. Low-quality specifications may hinder the learning process. We believe this limitation opens up exciting future work on AI-enabled specification generation. Moreover, the current version of HYPOLE is limited to discrete-action settings because it relies on value-based CTDE methods, which we choose over actor-critic methods due to their IGM-like guarantees. Finally, HYPOLE is not the method of choice when the task does not involve relational dependencies or temporal behaviors between agents and can instead be well captured by standard reward design, such as tasks with independent objectives or simple non-temporal requirements.

## Acknowledgment

This work is partially sponsored by the United States NSF Award SaTC 2245114.

## Impact Statement

This work takes a step toward bringing expressive specifications, in the form of hyperproperties, into practical MARL for partially observable and uncontrolled environments. By allowing tactics to be stated as precise mathematical formulas, HYPOLE can support the design of interpretable and verifiable behaviors in autonomous domains such as drone swarms. At the same time, the use of expressive specifications introduces potential risks. Malicious or adversarial specifications could be used to induce harmful behaviors, and poorly designed specifications may lead to unintended consequences. These risks highlight the importance of careful specification design, validation, and human oversight when deploying specification-guided MARL systems.

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

# A. Proofs

## A.1. Proof of Theorem 5.1

*Proof.* Let $\mathcal{M}$ be a POMDP and let $\mathbf{Skolem}(\varphi)$ denote the Skolemized form of the HyperLTL formula $\varphi$, written as

$$\mathbf{Skolem}(\varphi) = \exists \mathbf{f}_i(\tau_{i_1}, \dots, \tau_{i_{|\mathbb{Q}_i^{\forall}|}}) \ \forall \tau_j. \ \mathbf{Skolem}(\psi),$$

for each $i \in \mathbb{Q}^{\exists}$ and $j \in \mathbb{Q}^{\forall}$. Assume that the tuple of policies $\langle \pi_i^{\star} \rangle_{i \in \mathbb{Q}^{\exists}} \sqcup \langle \pi_j^{\star} \rangle_{j \in \mathbb{Q}^{\forall}}$ optimizes the robustness-based objective induced by $\mathbf{Skolem}(\varphi)$.

Since $\Phi$ is path-consistent, for any history $h$ generated by a latent path $\zeta$, $\hat{\zeta} \triangleq \arg\max_{\zeta \in \mathcal{Z}^*} \Phi(h, \zeta)$ coincides with the actual path $\zeta$ that generated $h$. Consequently, for any finite horizon $k \geq 0$, $\mathsf{Tr}(\hat{\zeta}_{[0:k]}) = \mathsf{Tr}(\zeta_{[0:k]})$.

Fix an arbitrary $k \geq 0$. The zipped path induces a trace assignment for the variables of $\varphi$ as follows.

- For each universally quantified trace variable $\tau_j$, where $j \in \mathbb{Q}^{\forall}$,

$$\tau_j \ \mapsto \ \mathsf{Tr}((\arg\max_{\zeta \in \mathcal{Z}^*} \Phi(h_j, \zeta))_{[0:k]}).$$

- For each existentially quantified trace variable $\tau_i$, where $i \in \mathbb{Q}^{\exists}$,

$$\tau_i \ \mapsto \ \mathbf{f}_i\Big( \mathsf{Tr}((\arg\max_{\zeta \in \mathcal{Z}^*} \Phi(h_{i_1}, \zeta))_{[0:k]}), \dots, \mathsf{Tr}((\arg\max_{\zeta \in \mathcal{Z}^*} \Phi(h_{i_{|\mathbb{Q}_i^{\forall}|}}, \zeta))_{[0:k]}) \Big).$$

By construction of Skolemization, each Skolem function $\mathbf{f}_i$ depends only on trace variables corresponding to universally quantified indices preceding $i$. Moreover, if $i_\ell = j$ for some $\ell$, then the corresponding trace prefixes coincide. Since the given policy tuple optimizes $\mathbf{Skolem}(\varphi)$, it maximizes the probability that the robustness value

$$\rho(\mathsf{zip}(\cdot), \mathbf{Skolem}(\psi))$$

converges to $\rho_{max}$. By the quantitative semantics of LTL, convergence of robustness to $\rho_{max}$ is equivalent to satisfaction of the inner LTL formula $\mathbf{Skolem}(\psi)$. Therefore, for the fixed horizon $k$, the induced trace assignment maximizes the probability of satisfying $\mathbf{Skolem}(\varphi)$.

**Lifting to the original HyperLTL formula.** Since Skolemization preserves satisfiability of HyperLTL formulas, and existentially quantified traces are instantiated as Skolem witnesses depending only on preceding universal traces, maximizing the robustness-based objective induced by $\mathbf{Skolem}(\varphi)$ yields a tuple of policies that maximizes the probability of satisfying the original HyperLTL formula $\varphi$ in $\mathcal{M}$. Hence, the tuple of policies $\langle \pi_i^{\star} \rangle_{i \in \mathbb{Q}^{\exists}} \sqcup \langle \pi_j^{\star} \rangle_{j \in \mathbb{Q}^{\forall}}$ optimizes the probability of satisfying $\varphi$ in $\mathcal{M}$. $\square$

## A.2. Proof of Theorem 5.3

*Proof.* Given a POMDP $\mathcal{M}$ and a HyperLTL formula $\varphi$, let $\mathbf{Skolem}(\varphi)$ be the Skolemized form of $\varphi$, and let the robustness values $\rho(\mathsf{zip}(\cdot), \mathbf{Skolem}(\psi))$ be used as the reward signal by the CTDE algorithm. The CTDE algorithm optimizes a joint action–value function $\mathbf{Q}(\mathbf{h}, \mathbf{a})$ that evaluates the expected discounted return under the robustness-based reward signal. Hence, an optimal joint policy $\boldsymbol{\pi}^{\star}$ computed as: $\mathbf{Q}^{\boldsymbol{\pi}^{\star}}(\mathbf{h}, \mathbf{a}) = \max_{\boldsymbol{\pi}} \mathbf{Q}^{\boldsymbol{\pi}}(\mathbf{h}, \mathbf{a})$,

We assume that $\mathbf{Q}$ satisfies the IGM property. Then there exist individual action–value functions $\langle Q_i \rangle_{\ell \in \{1, \dots, n\}}$ such that for every joint history $\mathbf{h} = \langle h_\ell \rangle_{\ell \in \{1, \dots, n\}}$,

$$\arg\max_{\mathbf{a}} \mathbf{Q}(\mathbf{h}, \mathbf{a}) = \Big\langle \arg\max_{a_\ell} Q_\ell(h_\ell, a_\ell) \Big\rangle_{\ell \in \mathbb{Q}^{\exists} \cup \mathbb{Q}^{\forall}}.$$

Hence we get optimal decentralized policies from:

$$Q_\ell^{\pi_\ell^{\star}}(h_\ell, a_\ell) = \max_{\pi} \ Q_\ell^{\pi_\ell}(h_\ell, a_\ell), \quad \text{for each } \ell \in \mathbb{Q}^{\exists} \cup \mathbb{Q}^{\forall}.$$

By the IGM equality above, the induced joint greedy action $\langle \pi_\ell^\star(h_\ell) \rangle_\ell$ attains a joint maximizer of $\mathbf{Q}(\mathbf{h}, \cdot)$ for given $\mathbf{h}$; therefore the induced joint policy is optimal w.r.t. $\mathbf{Q}$. It is important to note that we the input of the policies with the existential quantifier gets the history associated with

By construction, the CTDE algorithm uses the robustness values $\rho(\mathsf{zip}(\cdot), \mathbf{Skolem}(\psi))$ as the reward signal. Hence, the learned joint action–value function $\mathbf{Q}$ optimizes the same objective as (3), i.e., it maximizes (over tuples of policies) the probability that the robustness-value of the zipped traces w.r.t. $\mathbf{Skolem}(\psi)$ converges to $\rho_{\max}$. Therefore, any policy tuple induced by the decentralized policies $\langle \pi_i^\star \rangle_{i \in \mathbb{Q}^\exists}$ and $\langle \pi_j^\star \rangle_{j \in \mathbb{Q}^\forall}$ is an optimizer of (3), and thus optimizes the satisfaction probability of $\mathbf{Skolem}(\varphi)$. Note that our theoretical guarantees assume that, during decentralized execution, $f_{\text{enc-dec}}$ accurately reconstructs the relevant universally quantified trace prefixes required by the Skolem witnesses. $\qquad\square$

## B. Continuation of Problem Statement Running Example

Consider the POMDP in Fig. 2. We use the following HyperLTL formula:

$$\varphi_{\text{exp2}} \triangleq \forall \tau_1 \forall \tau_2. \ \mathsf{Dist}(\mathrm{pos}_{\tau_1}, \mathrm{pos}_{\tau_2}) < 3 \ \ \mathcal{U} \ \ (\mathrm{pos}_{\tau_1} = i)$$

FF, corresponding to $\tau_1$, samples histories using policy $\pi_1$, $\mathcal{H}_{\tau_1} = \{h_1^1, h_1^2\}$:

$$h_1^1 : \langle a \rangle \xrightarrow{\text{R}} \langle b \rangle \xrightarrow{\text{R}} \langle c, c \rangle \xrightarrow{\text{U}} \langle f, f, f \rangle \xrightarrow{\text{L}} \langle i, i \rangle$$
$$h_1^2 : \langle a \rangle \xrightarrow{\text{R}} \langle b \rangle \xrightarrow{\text{R}} \langle e \rangle \xrightarrow{\text{L}} \langle h \rangle \xrightarrow{\text{U}} \langle i, i \rangle$$

Med, corresponding to $\tau_2$, samples histories using policy $\pi_2$, $\mathcal{H}_{\tau_2} = \{h_2^1, h_2^2\}$:

$$h_2^1 : \langle a \rangle \xrightarrow{\text{U}} \langle d \rangle \xrightarrow{\text{U}} \langle g, g \rangle \xrightarrow{\text{R}} \langle d \rangle \xrightarrow{\text{R}} \langle d \rangle$$
$$h_2^2 : \langle a \rangle \xrightarrow{\text{U}} \langle b \rangle \xrightarrow{\text{L}} \langle e \rangle \xrightarrow{\text{R}} \langle f, f, f \rangle \xrightarrow{\text{L}} \langle i, i \rangle$$

Assume that $\Phi$ is path-consistent. Then we can construct the paths associated with $\mathcal{H}_{\tau_1}$ and $\mathcal{H}_{\tau_2}$ as $\mathcal{Z}_{\tau_1} = \{\arg\max_{\zeta \in \mathcal{Z}^*} \Phi(h_1^1, \zeta), \arg\max_{\zeta \in \mathcal{Z}^*} \Phi(h_1^2, \zeta)\}$ and $\mathcal{Z}_{\tau_2} = \{\arg\max_{\zeta \in \mathcal{Z}^*} \Phi(h_2^1, \zeta), \arg\max_{\zeta \in \mathcal{Z}^*} \Phi(h_2^2, \zeta)\}$, respectively. We now compute the probability of satisfying $\varphi_{\text{exp2}}$ using $\mathcal{Z}_{\tau_1}$ and $\mathcal{Z}_{\tau_2}$; the evaluation ranges over all pairwise combinations of paths in $\mathcal{Z}_{\tau_1} \times \mathcal{Z}_{\tau_2}$:

$$\mathsf{Traces}(\langle \{\arg\max_{\zeta \in \mathcal{Z}^*} \Phi(h_1^1, \zeta)\}, \{\arg\max_{\zeta \in \mathcal{Z}^*} \Phi(h_2^1, \zeta)\} \rangle) \not\models \varphi_{\text{exp2}} \quad \mathsf{Traces}(\langle \{\arg\max_{\zeta \in \mathcal{Z}^*} \Phi(h_1^1, \zeta)\}, \{\arg\max_{\zeta \in \mathcal{Z}^*} \Phi(h_2^2, \zeta)\} \rangle) \models \varphi_{\text{exp2}}$$

$$\mathsf{Traces}(\langle \{\arg\max_{\zeta \in \mathcal{Z}^*} \Phi(h_1^2, \zeta)\}, \{\arg\max_{\zeta \in \mathcal{Z}^*} \Phi(h_2^1, \zeta)\} \rangle) \models \varphi_{\text{exp2}} \quad \mathsf{Traces}(\langle \{\arg\max_{\zeta \in \mathcal{Z}^*} \Phi(h_1^2, \zeta)\}, \{\arg\max_{\zeta \in \mathcal{Z}^*} \Phi(h_2^2, \zeta)\} \rangle) \models \varphi_{\text{exp2}}$$

Out of the four possible pairs in $\mathcal{Z}_{\tau_1} \times \mathcal{Z}_{\tau_2}$, only three satisfy $\varphi_{\text{exp2}}$, yielding a satisfaction probability of $0.75$.

## C. More Technical Details for Sec. 5

### C.1. Details on Sec. 5.1

Let $\varphi = \mathbb{Q}_1 \tau_1 \ldots \mathbb{Q}_n \tau_n. \ \psi(\tau_1, \ldots, \tau_n)$ be a HyperLTL formula. The Skolemized formula has the form:

$$\mathbf{Skolem}(\varphi) = \underbrace{\exists \mathbf{f}_i(\tau_{i_1}, \ldots, \tau_{i_{|\mathbb{Q}_i^\forall|}})}_{\text{for each } i \in \mathbb{Q}^\exists}. \quad \underbrace{\forall \tau_j.}_{\text{for each } j \in \mathbb{Q}^\forall} \quad \mathbf{Skolem}(\psi)$$

Using the Transformation , we reformulate the problem in Fig. 3. We begin by defining the *image* of a Skolem function $\mathbf{f}_i$ as:

$$Img(\mathbf{f}_i) \triangleq \{\mathbf{f}_i(t_{i_1}, \ldots, t_{i_{|\mathbb{Q}_i^\forall|}}) \mid t_{i_j} \in \mathsf{Traces}(\bigcup_{h \in (\mathcal{H}_{i_j} \sim \mathcal{D}_{\pi_{i_j}})} \{\arg\max_{\zeta \in \mathcal{Z}^*} \Phi(h, \zeta)\}), \ j \in \mathbb{Q}_i^\forall\}$$

That is, $Img(\mathbf{f}_i)$ is the set of all traces obtained by mapping the preceding universally quantified traces $\tau_{i_j}$, where each $t_{i_j}$ ranges over the trace set derived from the corresponding history set $\mathcal{H}_{i_j}$. Now, let us use $\bowtie$ to indicate that the collection of

trace sets is ordered with respect to their path indices. That is, given two tuples of sets of traces $\mathcal{T}_1$ and $\mathcal{T}_2$, applying $\bowtie$ yields an ordered tuple $\mathcal{T}_1 \bowtie \mathcal{T}_2 \triangleq \langle \mathsf{Traces}(\mathcal{Z}_{\tau_x}) \rangle_{x \in \{1, \ldots, n\}}$, where each $\mathsf{Traces}(\mathcal{Z}_{\tau_x})$ belongs to either $\mathcal{T}_1$ or $\mathcal{T}_2$. Returning to the problem in Fig. 3, given a POMDP $\mathcal{M}$ and a HyperLTL formula $\varphi$, our goal is to compute: a tuple of Skolem witnesses $\langle \mathbf{f}_i \rangle_{i \in \mathbb{Q}^\exists}$, and a tuple of optimal policies $\langle \pi_j^\star \rangle_{j \in \mathbb{Q}^\forall}$, such that:

$$\langle \pi_j^\star \rangle_j \in \left[ \underset{\langle \pi_j \rangle}{\arg\max} \, \mathbb{P} \Big[ \langle Img(\mathbf{f}_i) \rangle \bowtie \langle \mathsf{Traces}( \bigcup_{h \in (\mathcal{H}_{i_j} \sim \mathcal{D}_{\pi_{i_j}})} \{ \underset{\zeta \in \mathcal{Z}^*}{\arg\max} \, \Phi(h, \zeta) \}) \rangle \models \mathbf{Skolem}(\varphi) \Big] \right]_{i \in \mathbb{Q}^\exists, j \in \mathbb{Q}^\forall}$$

The meaning of above formula is that the policy tuple $\langle \pi_j^\star \rangle$ maximizes the probability that the ordered collection consisting of (1) the generated traces of the universal quantifiers $\langle \mathsf{Traces}(\mathcal{Z}_{\tau_j}) \rangle_{j \in \mathbb{Q}^\forall}$ and (2) the Skolem witness for all existential quantifiers $\langle Img(\mathbf{f}_i) \rangle_{i \in \mathbb{Q}^\exists}$ jointly satisfies $\mathbf{Skolem}(\varphi)$. In the reformulated problem, policies are synthesized solely for universally quantified traces, whereas Skolem functions are learned for existentially quantified traces to serve as witnesses of optimality.

### C.2. Details on Sec. 5.2

We reformulate the HyperLTL satisfaction problem as an optimization problem under its quantitative semantics, where the Skolemized formula $\mathbf{Skolem}(\varphi)$ is evaluated on tuples of histories $\langle h_1, \ldots, h_n \rangle$ sampled from the POMDP $\mathcal{M}$.

**Robustness for a Single Trace.** Let $\mathbb{R}$ denote the set of real numbers and let $\Psi$ denote the set of all LTL formulas. Let $f : 2^{\mathsf{AP}} \to \mathbb{R}$ be a valuation function that assigns real values to sets of atomic propositions. Let $\mathbb{R}$ be the set of real numbers, $\Psi$ the set of all LTL formulas, and $f : 2^{\mathsf{AP}} \to \mathbb{R}$ be a *valuation function* assigning real values to sets of atomic propositions. For a state $s \in S$ of $\mathcal{M}$, quantitative semantics are defined by predicates of the form $f(L(s)) < c$, where $c$ is a user-specified threshold (see Fig. 9). Different values of $c$ can be assigned to different segments of the inner LTL formula to emphasize the relative priority of those segments. The robustness function $\rho : \mathsf{Traces}(\mathcal{Z}^*) \times \Psi \to \mathbb{R}$ assigns a real-valued score to a finite trace and an LTL formula, bounded by $\rho_{max}$ and $\rho_{min}$. For a given trace, higher $\rho$ values correspond to greater robustness in satisfying $\psi$, while lower values indicate weaker satisfaction or possible violation. Formally, given an LTL formula $\psi$ and a POMDP $\mathcal{M}$, we formulate the problem of synthesizing a policy $\pi^\star$ as: Formally, given an LTL formula $\psi$ and a POMDP $\mathcal{M}$, we formulate the policy synthesis problem as:

$$\pi^\star \in \underset{\pi}{\arg\max} \, \underset{h \sim \mathcal{D}_\pi}{\mathbb{P}} \left[ \rho \big( \mathsf{Tr}((\underset{\zeta \in \mathcal{Z}^*}{\arg\max} \, \Phi(h, \zeta))_{[0:k]}), \psi \big) \overset{\star}{\to} \rho_{max} \right],$$

where $\overset{\star}{\to}$ means convergence. I.e., $\pi^\star$ maximizes the probability that paths induced by histories under $\pi$ satisfy $\psi$.

**Robustness for a Tuple of Traces.** Next, to compute robustness over multiple (universally quantified) traces, we define a $\mathsf{zip}$ operator as follows: that performs pointwise aggregation over a tuple of traces. Given a tuple of finite traces $\langle t_\ell \rangle_{\ell \in \{1, \ldots, n\}}$, where each $t_\ell = t_\ell(0) t_\ell(1) \cdots t_\ell(m_\ell)$, we define their *zipped trace* as:

$$\mathsf{zip}(\langle t_\ell \rangle_{\ell \in \{1, \ldots, n\}}) \triangleq \langle t_\ell(0) \rangle_{\ell \in \{1, \ldots, n\}} \cdots \langle t_\ell(k) \rangle_{\ell \in \{1, \ldots, n\}}$$

Given an LTL formula $\psi$, a tuple of paths $\langle \zeta_1, \zeta_2, \ldots, \zeta_n \rangle$ is more likely to satisfy $\psi$ if the robustness value of $\mathsf{zip}(\langle \mathsf{Tr}(\zeta_{1[0:k_1]}), \mathsf{Tr}(\zeta_{2[0:k_2]}), \ldots, \mathsf{Tr}(\zeta_{n[0:k_n]}) \rangle)$ converges to $\rho_{max}$ with respect to $\psi$ for some $k_1, \ldots, k_n$, where $0 \le k_\ell \le |\zeta_\ell|$ for each $1 \le \ell \le n$.

Thus, the optimization problem of computing a tuple of policies $\langle \pi_1^\star, \pi_2^\star, \ldots, \pi_n^\star \rangle$ that maximizes robustness can be stated as:

$$\langle \pi_\ell^\star \rangle_\ell \in \left[ \underset{\langle \pi_\ell \rangle}{\arg\max} \, \mathbb{P} \left[ \rho \big( \mathsf{zip} \langle \mathsf{Tr}((\underset{\zeta \in \mathcal{Z}^*}{\arg\max} \, \Phi(h_\ell \sim \mathcal{D}_{\pi_\ell}, \zeta))_{[0:k_l]}) \rangle \big), \psi \big) \overset{\star}{\to} \rho_{max} \right] \right]_{\ell \in \{1, \ldots n\}}$$

This transformation highlights that LTL specifications are implicitly universally quantified in the single-trace setting.

**Robustness for Skolemized HyperLTL.** We now address the optimization problem induced by an alternating HyperLTL formula. Let $\langle t_i \rangle_{i \in I}$ and $\langle t_j \rangle_{j \in J}$ be two trace tuples, where $I \cup J = \{1, \ldots, n\}$ and $I \cap J = \emptyset$. To preserve index order when combining them, we use $\sqcup$, which produces a single tuple ordered by path indices. In particular, applying $\sqcup$ produces

$$
\begin{aligned}
\rho\big(\mathsf{Tr}(\zeta_{[\ell:k]}),\psi\big) &= \rho_{min} \text{ if } \mathsf{Tr}(\zeta_{[\ell:\cdot]}) = \epsilon \text{ and } \rho\big(\mathsf{Tr}(\zeta_{[\ell:k]}),\psi\big) \text{ otherwise.} \\
\rho\big(\mathsf{Tr}(\zeta_{[\ell:k]}),\mathsf{true}\big) &= \rho_{max} \\
\rho\big(\mathsf{Tr}(\zeta_{[\ell:k]}),f\big(L(s_\ell) < c\big)\big) &= c - f\big(L(s_\ell)\big) \\
\rho\big(\mathsf{Tr}(\zeta_{[\ell:k]}),\neg\psi\big) &= -\rho\big(\mathsf{Tr}(\zeta_{[\ell:k]}),\psi\big) \\
\rho\big(\mathsf{Tr}(\zeta_{[\ell:k]}),\bigcirc\psi\big) &= \rho\big(\mathsf{Tr}(\zeta_{[\ell+1:k]}),\psi\big) \text{ if } (k > \ell). \\
\rho\big(\mathsf{Tr}(\zeta_{[\ell:k]}),\square\psi\big) &= \min_{i\in[\ell,k)}\rho\big(\mathsf{Tr}(\zeta_{[i:k]}),\psi\big) \\
\rho\big(\mathsf{Tr}(\zeta_{[\ell:k]}),\lozenge\psi\big) &= \max_{i\in[\ell,k)}\rho\big(\mathsf{Tr}(\zeta_{[i:k]})\psi\big) \\
\rho\big(\mathsf{Tr}(\zeta_{[\ell:k]}),\psi_1\wedge\psi_2\big) &= \min\big(\rho\big(\mathsf{Tr}(\zeta_{[\ell:k]}),\psi_1\big)\rho\big(\mathsf{Tr}(\zeta_{[\ell:k]}),\psi_2\big)\big) \\
\rho\big(\mathsf{Tr}(\zeta_{[\ell:k]}),\psi_1\vee\psi_2\big) &= \max\big(\rho\big(\mathsf{Tr}(\zeta_{[\ell:k]}),\psi_1\big)\rho\big(\mathsf{Tr}(\zeta_{[\ell:k]}),\psi_2\big)\big) \\
\rho\big(\mathsf{Tr}(\zeta_{[\ell:k]}),\psi_1\,\mathcal{U}\,\psi_2\big) &= \max_{i\in[\ell,k)}\Big(\min\Big(\rho\big(\mathsf{Tr}(\zeta_{[i:k]}),\psi_2\big), \min_{j\in[\ell,i)}\rho\big(\mathsf{Tr}(\zeta_{[j:i]}),\psi_1\big)\Big)\Big)
\end{aligned}
$$

**Figure 9.** Quantitative semantics for LTL, adapted from (Hsu et al., 2025; Li et al., 2017).

a tuple $\mathsf{Tr}(\zeta_1) < \mathsf{Tr}(\zeta_2) < \cdots < \mathsf{Tr}(\zeta_n)$, where $\mathsf{Tr}(\zeta_x) < \mathsf{Tr}(\zeta_y)$ denotes $x < y$. Based on this construction, satisfaction of the inner LTL body $\psi$ is defined over tuples of histories. For each $i \in \mathbb{Q}^\exists$ and $j \in \mathbb{Q}^\forall$, a tuple $\langle h_1, \ldots, h_n \rangle$ satisfies $\psi$ if and only if:

$$
\rho\Big(\mathsf{zip}\big(\langle\mathsf{Tr}(\arg\max_{\zeta\in\mathcal{Z}^*}\Phi(h_i,\zeta))_{[0:k_i]}\rangle\big) \sqcup \langle\mathsf{Tr}(\arg\max_{\zeta\in\mathcal{Z}^*}\Phi(h_j,\zeta))_{[0:k_j]}\rangle\rangle),\mathbf{Skolem}(\psi)\Big) \xrightarrow{\star} \rho_{max}
$$

Therefore, we formulate the optimization problem for the Skolemized HyperLTL formula as follows:

$$
\langle\pi_i^\star\rangle \sqcup \langle\pi_j^\star\rangle \in \Bigg[\arg\max_{\langle\pi_i\rangle\sqcup\langle\pi_j\rangle}\mathbb{P}\Big[\rho\Big(\mathsf{zip}\big(\langle\mathsf{Tr}\big((\arg\max_{\zeta\in\mathcal{Z}^*}\Phi(h_i\sim\mathcal{D}_{\pi_i},\zeta))_{[0:k_i]}\big)\rangle \sqcup
$$
$$
\langle\mathsf{Tr}\big((\arg\max_{\zeta\in\mathcal{Z}^*}\Phi(h_j\sim\mathcal{D}_{\pi_j},\zeta))_{[0:k_j]}\big)\rangle\big)\big),\mathbf{Skolem}(\psi)\Big) \xrightarrow{\star} \rho_{max}\Big]\Bigg]_{i\in\mathbb{Q}^\exists,j\in\mathbb{Q}^\forall}
$$

## D. Experimental Setup and Implementation Details

In this section, we detail the implementation, architecture, and hyperparameters used in our learning process.

### D.1. Details on MARL used in HYPOLE

In our experiments, we use the implementations of QMIX, VDN, QTRAN, and IQL provided by https://github.com/oxwhirl/pymarl. For QMIX, we set `epsilon_start=1.0`, `epsilon_finish=0.05`, `epsilon_anneal_time=50000`, `buffer_size=5000`, `target_update_interval=200`, `mixing_embed_dim=64`, `opt_loss=1`, and `nopt_min_loss=0.1`. For VDN, we set `epsilon_start=1.0`, `epsilon_finish=0.05`, `epsilon_anneal_time=50000`, `buffer_size=5000`, and `target_update_interval=200`. For QTRAN, we set `epsilon_start=1.0`, `epsilon_finish=0.05`, `epsilon_anneal_time=50000`, `buffer_size=5000`, `target_update_interval=200`, `mixing_embed_dim=64`, `opt_loss=1`, and `nopt_min_loss=0.1`. Finally, for IQL, we set `epsilon_start=1.0`, `epsilon_finish=0.05`, `epsilon_anneal_time=50000`, `buffer_size=5000`, and `target_update_interval=200`.

**Table 1.** SMAC maps used in evaluation, including unit configurations and HyperLTL formulas used in our experiments.

| Map Name | Ally Units | Enemy Units | Formula | Quantifiers |
|---|---|---|---|---|
| 3m | 3 Marines | 3 Marines | $\varphi_{\text{focus-fire}}, \varphi_{\text{defense}}, \varphi_{\text{bad}}$ | $\forall^*$ |
| 8m | 3 Marines | 3 Marines | $\varphi_{\text{focus-fire}}$ | $\forall^*$ |
| 2s3z | 2 Stalkers & 3 Zealots | 2 Stalkers & 3 Zealots | $\varphi_{\text{focus-fire}}$ | $\forall^*$ |
| 3s5z | 3 Stalkers & 5 Zealots | 3 Stalkers & 5 Zealots | $\varphi_{\text{focus-fire}}$ | $\forall^*$ |
| MMM | 1 Med & 2 Marauders & 7 Marines | 1 Med & 2 Marauders & 7 Marines | $\varphi_{\text{medivac}}$ | $\forall^*, \forall^*\exists$ |
| 5m_vs_6m | 5 Marines | 6 Marines | $\varphi_{\text{focus-fire}}, \varphi_{\text{kite}}$ | $\forall^*$ |
| MMM2 | 1 Med, 2 Marauders & 7 Marines | 1 Med, 2 Marauders & 8 Marines | $\varphi_{\text{medivac}}$ | $\forall^*, \forall^*\exists$ |
| bane_vs_bane | 20 Zerglings & 4 Banelings | 20 Zerglings & 4 Banelings | $\varphi_{\text{focus-fire}}$ | $\forall^*$ |
| corridor | 6 Zealots | 24 Zerglings | $\varphi_{\text{corridor}}$ | $\forall^*$ |

## D.2. Details on $f_{\texttt{enc-dec}}, \mathbf{f}, \Omega$ used in HYPOLE

$f_{\texttt{enc-dec}}$ is a feedforward `encoder-decoder` (autoencoder) model implemented in PyTorch. Both the encoder and decoder are three-layer MLPs composed of `Linear` layers. We use `ReLU` activations after each hidden layer, and apply a final `Tanh` activation at the decoder output to bound predictions to $[-1, 1]$, matching the normalized target range. The first hidden layer has width equal to the observation dimension, and the second hidden layer has 256 units; the remaining layer sizes follow this encoder–decoder symmetry. We optimize the model using Adam with learning rate $10^{-3}$ and train it by minimizing mean squared error (MSE).

To train $f_{\texttt{enc-dec}}$ model, we maintain a buffer $\mathcal{D}$ containing the 10,000 most recent agent with existential quantifier observations and their corresponding global states at the same timestep. After each episode, we uniformly sample a mini-batch of size 32 from the pair buffer and perform one gradient update step. To reduce unnecessary computation, we apply a decaying update schedule based $\beta$ on the current training loss. When the loss exceeds $9 \times 10^{-5}$, we update $f_{\texttt{enc-dec}}$ every 50 environment steps. When the loss is between $9 \times 10^{-5}$ and $9 \times 10^{-6}$, we update every 100 steps. Once the loss drops below $9 \times 10^{-6}$, we update every 1000 steps.

Next, for each agent with an existential quantifier $i \in \mathbb{Q}^{\exists}$, we use an `RNN` to approximate $\Omega(\mathbf{f}_i(\cdot))$ and obtain a history representation $\hat{h}_i$. We then use $\hat{h}_i$ to construct the agent's utility function and decentralized policy $\pi_i$. For the remaining components of CTDE, we follow the standard PyMARL implementation. It is important to note that the `RNN` approximating $\Omega(\mathbf{f}_i(\cdot))$ is trained jointly with CTDE.

## D.3. Details on Experimental Setup and Training

We ran training on three machines. The first machine has an AMD EPYC 7742 64-core CPU and three NVIDIA RTX A6000 GPUs. The second is a shared server with an Intel Xeon Platinum 8268 CPU and four NVIDIA Quadro RTX 8000 GPUs; on this machine, CPU resources are limited due to sharing (GPU resources are not). The third machine has an AMD EPYC 7742 64-core CPU and three NVIDIA RTX PRO 6000 Blackwell Max-Q Workstation Edition GPUs. For SMAC and MessySMAC, we train each agent for 10k environment steps and evaluate every checkpoint over 32 test episodes. All reported SMAC and MessySMAC results are obtained in `test_mode`. For WildFire, we train each agent for 5k environment steps and evaluate every checkpoint over 30 test episodes. All SMAC and Wildfire experiments were conducted over ten independent runs with different random seeds, while MessySMAC experiments were conducted over five independent runs.

## E. Details on SMAC and MessySMAC Benchmark

We evaluate on the StarCraft II Multi-Agent Challenge (SMAC) benchmark https://github.com/oxwhirl/smac. Table 1 lists the scenarios used in our experiments, together with the HyperLTL formulas we evaluate and their corresponding quantifier structures. For baseline comparisons, we use the shaped reward provided by SMAC, which combines hit-point damage, enemy kills, and an additional bonus for winning the battle. SMAC also includes a sparse reward ($+1$ for winning and $-1$ for losing an episode); however, in our preliminary experiments the sparse reward performed substantially worse than the shaped reward. Therefore, we use the shaped reward for all baseline results.

We mostly use the same configuration for MessySMAC. However, this benchmark adds observation stochasticity $\phi$ and

**Table 2.** WildFire maps used in evaluation, including agents configurations and HyperLTL formulas used in our experiments.

| Map Name | Grid Size | Agents | Objects | Formula |
|---|---|---|---|---|
| 5P_1F1V | $5 \times 5$ | 1 FF, 1 Med | 1 Victim 1 Fire Zone | $\varphi_{\text{wildfire}}$ |
| 5P_3F2V | $5 \times 5$ | 1 FF, 1 Med | 2 Victim 3 Fire Zone | $\varphi_{\text{wildfire}}$ |
| 8P_3F2V | $8 \times 8$ | 1 FF, 1 Med | 2 Victim 3 Fire Zone | $\varphi_{\text{wildfire}}$ |
| 10P_3F2V | $10 \times 10$ | 1 FF, 1 Med | 2 Victim 3 Fire Zone | $\varphi_{\text{wildfire}}$ |

randomized initialization with $K$ random steps before each episode starts. For the 3m map, we use $(K = 10, \phi = 0.15)$. For the 8m, 2s3z, 3s5z, and MMM maps, we use $(K = 12, \phi = 0.15)$.

## F. Details on Wildfire Benchmark

We extend the Wildfire scenario from Hsu et al. (2025) to a partially observable setting. The environment contains two agent types: a firefighter drone FF 🧯 and a medical drone Med 🚁 (see Fig. 2). The agents must extinguish fires and rescue victims within a fixed time while maintaining a safe separation distance. In this setting, each agent observes only entities within a fixed sight range. We use the maps listed in Table 2. The global state includes all agent features and object (fire/victim) features, the last action, and the time step. Agent and object features are represented by centered coordinates: $(x - \text{center}_x, y - \text{center}_y)$, normalized by the maximum horizontal and vertical distances, respectively. The last-action vector encodes each agent's previous action. Each agent's observation consists of its move features, own features, the time step, and the features of agents and objects within its sight range. Move features specify the set of available actions. The action space is $\{\text{Up}, \text{Down}, \text{Left}, \text{Right}\}$. For each observed entity, we include relative position (normalized by sight range), distance to the entity (normalized by sight range), the entity type ID, and their last action.

We introduce three reasonable baseline reward functions $R_{\text{Wild\_1}}$, $R_{\text{Wild\_2}}$, $R_{\text{Wild\_3}}$. The functions are as follows:

$$R_{\text{Wild\_1}} = \begin{cases} 20 & \text{Victim Saved} \\ 20 & \text{Fire Ext} \\ -10 & \text{Out of Range} \\ -10 & \text{Med in Fire} \end{cases} \quad R_{\text{Wild\_2}} = \begin{cases} 20 & \text{Victim Saved} \\ 20 & \text{Fire Ext} \\ -10 & \text{Out of Range} \\ -10 & \text{Med in Fire} \\ +100 & \text{Mission Accomplish} \end{cases} \quad R_{\text{Wild\_3}} = \begin{cases} 20 & \text{Victim Saved} \\ 20 & \text{Fire Ext} \\ -10 & \text{Out of Range} \\ -10 & \text{Med in Fire} \\ +100 & \text{Mission Accomplish} \\ -50 & \text{Mission failed} \end{cases}$$

## G. HyperLTL Formulas

We provide all HyperLTL formulas used in the SMAC and MessySMAC experiments, namely $\varphi_{\text{focus-fire}}$, $\varphi_{\text{medivac}}$, $\varphi_{\text{kite}}$, $\varphi_{\text{corridor}}$, $\varphi_{\text{defense}}$, and $\varphi_{\text{bad}}$, as well as $\varphi_{\text{wildfire}}$ for the WildFire benchmark.

$$\varphi_{\text{focus-fire}} \triangleq \overbrace{\forall \tau_1, \ldots, \tau_n}^{\text{Ally Agents}} . \left[ \left( \left( \overbrace{(\text{dist}(\text{pos}_{\tau_1}, \text{pos\_enm\_1}) < \text{s\_range}_{\tau_1} \wedge \ldots \wedge \text{dist}(\text{pos}_{\tau_n}, \text{pos\_enm\_1}) < \text{s\_range}_{\tau_n})}^{\text{Focused Shooting}} \right. \right. \right.$$

$$\left. \cdots \vee \left( \text{dist}(\text{pos}_{\tau_1}, \text{pos\_enm\_m}) < \text{s\_range}_{\tau_1} \wedge \ldots \wedge \text{dist}(\text{pos}_{\tau_n}, \text{pos\_enm\_m}) < \text{s\_range}_{\tau_n} \right) \right) \wedge$$

$$\left( \underbrace{(\text{health\_enm\_1} < \text{health\_enm\_prev\_1}) \vee \cdots \vee (\text{health\_enm\_m} < \text{health\_enm\_prev\_m})}_{\text{Damaging Enemies Health/Shield Bar}} \right)$$

$$\mathcal{U} \underbrace{\text{\#enm\_dead} > \text{\#enm\_dead\_prev}}_{\text{Eliminating Enemies}} \left. \right] \mathcal{U} \text{ Win}$$

$$\varphi_{\text{medivac}} \triangleq \overbrace{\forall \tau_1, \ldots, \tau_{n-1}}^{\text{Ally Agents}} \overbrace{\forall \tau_n / \exists \tau_n}^{\text{Medivac}} .$$

$$\left[ \left( \left( \overbrace{\left( (\mathsf{dist}(\mathsf{pos}_{\tau_1}, \mathsf{pos\_enm\_1}) < \mathsf{s\_range}_{\tau_1} \wedge \ldots \wedge \mathsf{dist}(\mathsf{pos}_{\tau_{n-1}}, \mathsf{pos\_enm\_1}) < \mathsf{s\_range}_{\tau_{n-1}}) \vee}^{\text{Focused Shooting}} \right. \right. \right.$$

$$\left. \cdots \vee \left( \mathsf{dist}(\mathsf{pos}_{\tau_1}, \mathsf{pos\_enm\_m}) < \mathsf{s\_range}_{\tau_1} \wedge \ldots \wedge \mathsf{dist}(\mathsf{pos}_{\tau_{n-1}}, \mathsf{pos\_enm\_m}) < \mathsf{s\_range}_{\tau_{n-1}}) \right) \wedge \right.$$

$$\overbrace{\left( \mathsf{dist}(\mathsf{pos}_{\tau_n}, \mathsf{Min\_Health}(\tau_1, \ldots, \tau_{n-1})) < \mathsf{h\_range}_{\tau_n} \right)}^{\text{Medivac Close to Agent With Min Health}} \wedge$$

$$\underbrace{\left( \left( \sum_{i=1}^{n-1} \mathsf{health\_prev}_{\tau_i} - \sum_{i=1}^{n-1} \mathsf{health}_{\tau_i} \right) < \left( \sum_{i=1}^{m} \mathsf{health\_enm\_prev\_i} - \sum_{i=1}^{m} \mathsf{health\_enm\_i} \right) \right)}_{\text{Damaging Enemies Health/Shield Bar While Preserving Allies Health/Shield Bar}}$$

$$\mathcal{U} \underbrace{\texttt{\#enm\_dead} > \texttt{\#enm\_dead\_prev}}_{\text{Eliminating Enemies}} \bigg] \ \mathcal{U} \ \texttt{Win}$$

$$\varphi_{\text{kite}} \triangleq \overbrace{\forall \tau_1, \ldots, \tau_n}^{\text{Ally Agents}} . \left[ \left( \overbrace{\left( \neg \texttt{ready}_{\tau_1} \vee \left( \bigwedge_{i=1}^{m} \mathsf{dist}(\mathsf{pos}_{\tau_1}, \mathsf{pos\_enm\_i}) < \mathsf{s\_range\_i} \right) \right) \wedge}^{\text{Kiting}} \right. \right.$$

$$\left. \ldots \wedge \left( \neg \texttt{ready}_{\tau_n} \vee \left( \bigwedge_{i=1}^{m} \mathsf{dist}(\mathsf{pos}_{\tau_n}, \mathsf{pos\_enm\_i}) < \mathsf{s\_range\_i} \right) \right) \wedge \right.$$

$$\left( \underbrace{(\texttt{health\_enm\_1} < \texttt{health\_enm\_prev\_1})}_{\text{Damaging Enemies Health/Shield Bar}} \vee \cdots \vee (\texttt{health\_enm\_m} < \texttt{health\_enm\_prev\_m}) \right)$$

$$\mathcal{U} \underbrace{\texttt{\#enm\_dead} > \texttt{\#enm\_dead\_prev}}_{\text{Eliminating Enemies}} \bigg] \ \mathcal{U} \ \texttt{Win}$$

$$\varphi_{\text{corridor}} \triangleq \overbrace{\forall \tau_1, \ldots, \tau_n}^{\text{Ally Agents}} . \left[ \left( \overbrace{\mathsf{dist}(\mathsf{pos}_{\tau_1}, \mathsf{pos\_choke\_point}) < \delta \wedge \ldots \wedge \mathsf{dist}(\mathsf{pos}_{\tau_n}, \mathsf{pos\_choke\_point}) < \delta}^{\text{Defend in Choke Point}} \right) \mathcal{U} \right.$$

$$\left( \underbrace{(\texttt{health\_enm\_1} < \texttt{health\_enm\_prev\_1})}_{\text{Damaging Enemies Health/Shield Bar}} \vee \cdots \vee (\texttt{health\_enm\_m} < \texttt{health\_enm\_prev\_m}) \right)$$

$$\mathcal{U} \underbrace{\texttt{\#enm\_dead} > \texttt{\#enm\_dead\_prev}}_{\text{Eliminating Enemies}} \bigg] \ \mathcal{U} \ \texttt{Win}$$

$$\varphi_{\text{defense}} \triangleq \overbrace{\forall \tau_1, \ldots, \tau_n}^{\text{Ally Agents}} . \square \left( \overbrace{\bigwedge_{i=1}^{n} \bigwedge_{j=1}^{m} (\mathsf{dist}(\mathsf{pos}_{\tau_i}, \mathsf{pos\_enm\_j}) > \mathsf{o\_range\_enm\_j})}^{\text{Running Away from Enemies}} \right) \wedge \square \left( \underbrace{\bigwedge_{i=1}^{n} (\texttt{health}_{\tau_i} > \delta)}_{\text{Preserve Allies Health/Shield Bar}} \right)$$

$$\varphi_{\text{bad}} \triangleq \overbrace{\forall \tau_1, \ldots, \tau_n}^{\text{Ally Agents}} . \left[ \left( \overbrace{\left( (\mathsf{dist}(\mathsf{pos}_{\tau_1}, \mathsf{pos\_enm\_1}) < \mathsf{s\_range}_{\tau_1} \wedge \ldots \wedge \mathsf{dist}(\mathsf{pos}_{\tau_n}, \mathsf{pos\_enm\_1}) < \mathsf{s\_range}_{\tau_n}) \vee}^{\text{Focused Shooting}} \right. \right.$$

$$\left. \cdots \vee \left( \mathsf{dist}(\mathsf{pos}_{\tau_1}, \mathsf{pos\_enm\_m}) < \mathsf{s\_range}_{\tau_1} \wedge \ldots \wedge \mathsf{dist}(\mathsf{pos}_{\tau_n}, \mathsf{pos\_enm\_m}) < \mathsf{s\_range}_{\tau_n}) \right) \mathcal{U} \right.$$

$$\underbrace{\texttt{\#enm\_dead} > \texttt{\#enm\_dead\_prev}}_{\text{Eliminating Enemies}} \bigg] \ \mathcal{U} \ \texttt{Win}$$

$$\varphi_{\text{wildfire}} \triangleq \overbrace{\forall \tau_1}^{\text{Med}} . \underbrace{(\forall \tau_2 / \exists \tau_2)}_{\text{FF}} . \left[ \left( \left( \left( \overbrace{(\mathsf{dist}(\mathsf{pos}_{\tau_2}, \mathsf{pos\_fire\_1}) < \mathsf{prev\_dist\_FF\_fire\_1}_{\tau_2})}^{\text{FF Get Close to Fire}} \right) \vee \right. \right. \right.$$

$$\left. \ldots \vee (\mathsf{dist}(\mathsf{pos}_{\tau_2}, \mathsf{pos\_fire\_n}) < \mathsf{prev\_dist\_fire\_n}_{\tau_2}) \right) \mathcal{U} \, \mathsf{Ex\_fire}_{\tau_2} \right) \wedge$$

$$\left( \left( (\mathsf{dist}(\mathsf{dist}(\mathsf{pos}_{\tau_1}, \mathsf{pos\_Vic\_1}) < \mathsf{prev\_dist\_Vic\_1}_{\tau_1}) \vee \right. \right.$$

$$\left. \ldots \vee (\mathsf{dist}(\mathsf{dist}(\mathsf{pos}_{\tau_1}, \mathsf{pos\_Vic\_m}) < \mathsf{prev\_dist\_Vic\_m}_{\tau_1}) \right) \mathcal{U} \, \mathsf{Save\_Vic}_{\tau_1} \right) \right) \wedge$$

$$\mathsf{dist}(\mathsf{pos}_{\tau_1}, \mathsf{pos}_{\tau_2}) < \delta \ \wedge \ \bigwedge_{i=0}^{n} \neg \mathsf{pos\_Fire\_i}_{\tau_1} \right] \mathcal{U} \, \mathsf{Mission\_Accomplished}$$

## H. Additional Results

### H.1. HYPOLE vs. Shaped Rewards

In addition to the maps discussed in Sec. 6, we provide results on additional SMAC scenarios. Fig. 13 summarizes performance on 3m, 2s3z, and MMM2. As shown in Fig. 13a, 3m is relatively easy and all methods achieve similar performance. In contrast, on 2s3z (Fig. 13b) we observe clear gains for HYPOLE +QTRAN (brown line) and HYPOLE +IQL (gray line) over vanilla QTRAN (purple line) and IQL (pink line). On MMM2 (Fig. 13c), where we apply $\varphi_{\text{medivac}}$ within HYPOLE, HYPOLE +QMIX (orange line) clearly outperforms vanilla QMIX (blue line). HYPOLE +IQL (gray line) also achieves a non-trivial win rate (around $5\%$), whereas vanilla IQL (pink line) attains $0\%$. While both HYPOLE +QTRAN and vanilla QTRAN obtain $0\%$ win rate, HYPOLE +QTRAN (brown line) eliminates more enemies than vanilla QTRAN (see Fig. 17i).

The remaining MessySMAC maps are shown in Fig. 14. In the 3m map Fig. 14a, HYPOLE +MARL substantially outperforms the vanilla MARL baselines across all cases. In the 2s3z map Fig. 14b, HYPOLE +QMIX (orange line) and HYPOLE +VDN (red line) substantially outperform vanilla QMIX (blue line) and vanilla VDN (green line), respectively.

To better illustrate the significance of HYPOLE, we report its success-rate gains across all SMAC maps in Fig. 10. HYPOLE +QMIX (blue line) outperforms vanilla QMIX on almost all maps, except 5m_vs_6m and 3s5z; however, on 3s5z, it still surpasses the baseline in the final training steps. On hard scenarios such as MMM2 Fig. 10g, bane_vs_bane Fig. 10h, and corridor Fig. 10i, HYPOLE +QMIX achieves substantial improvements, reaching up to a $90\%$ gain in success rate on bane_vs_bane. Similarly, HYPOLE +VDN (orange line) improves performance in most cases, except on 2s3z, where the behavior is more unstable, while still achieving substantial gains on bane_vs_bane. Finally, HYPOLE +QTRAN (green line) outperforms vanilla QTRAN in nearly all scenarios except 5m_vs_6m, and achieves notable improvements on MMM, with up to a $50\%$ gain Fig. 10f. In addition to CTDE methods, HYPOLE +IQL (red line) performs well on almost all maps, achieving about a $60\%$ gain on 8m Fig. 10d.

We perform the same analysis for MessySMAC to further illustrate the significance of HYPOLE, and report its success-rate gains across all MessySMAC maps in Fig. 11. HYPOLE +QMIX (blue line) improves performance on most maps, especially on 3m Fig. 11a, where it achieves up to a $50\%$ gain. HYPOLE +VDN (orange line) improves the win rate over vanilla VDN in all cases, and performs particularly well on the harder MMM map Fig. 11e, with up to a $30\%$ gain. HYPOLE +QTRAN (green line) performs better in most cases, except on MMM Fig. 11e; however, it shows substantial improvement on 3m Fig. 11a, with up to a $50\%$ gain. Similarly, HYPOLE +IQL (red line) performs better on most maps, especially on 3m Fig. 11a, with up to a $\approx 35\%$ gain, except on 8m Fig. 11d, where neither HYPOLE +IQL nor vanilla IQL achieves any wins.

We also report the number of dead enemies and dead allies across all SMAC maps in Figs. 16 and 17 and all MessySMAC maps in Figs. 18 and 19.

In Fig. 12, we report results on the WildFire scenarios, focusing on the number of steps required to save victims and extinguish fires. On the 5P_1F1V map (Figs. 12a to 12c), HYPOLE +MARL consistently outperforms the vanilla baselines, requiring fewer steps to complete the objectives. On the 5P_3F2V map (Figs. 12d to 12f), HYPOLE +MARL outperforms the vanilla baselines in most cases, except that QTRAN+$R_{\text{Wild\_2}}$ and QTRAN+$R_{\text{Wild\_3}}$ (purple line) achieve

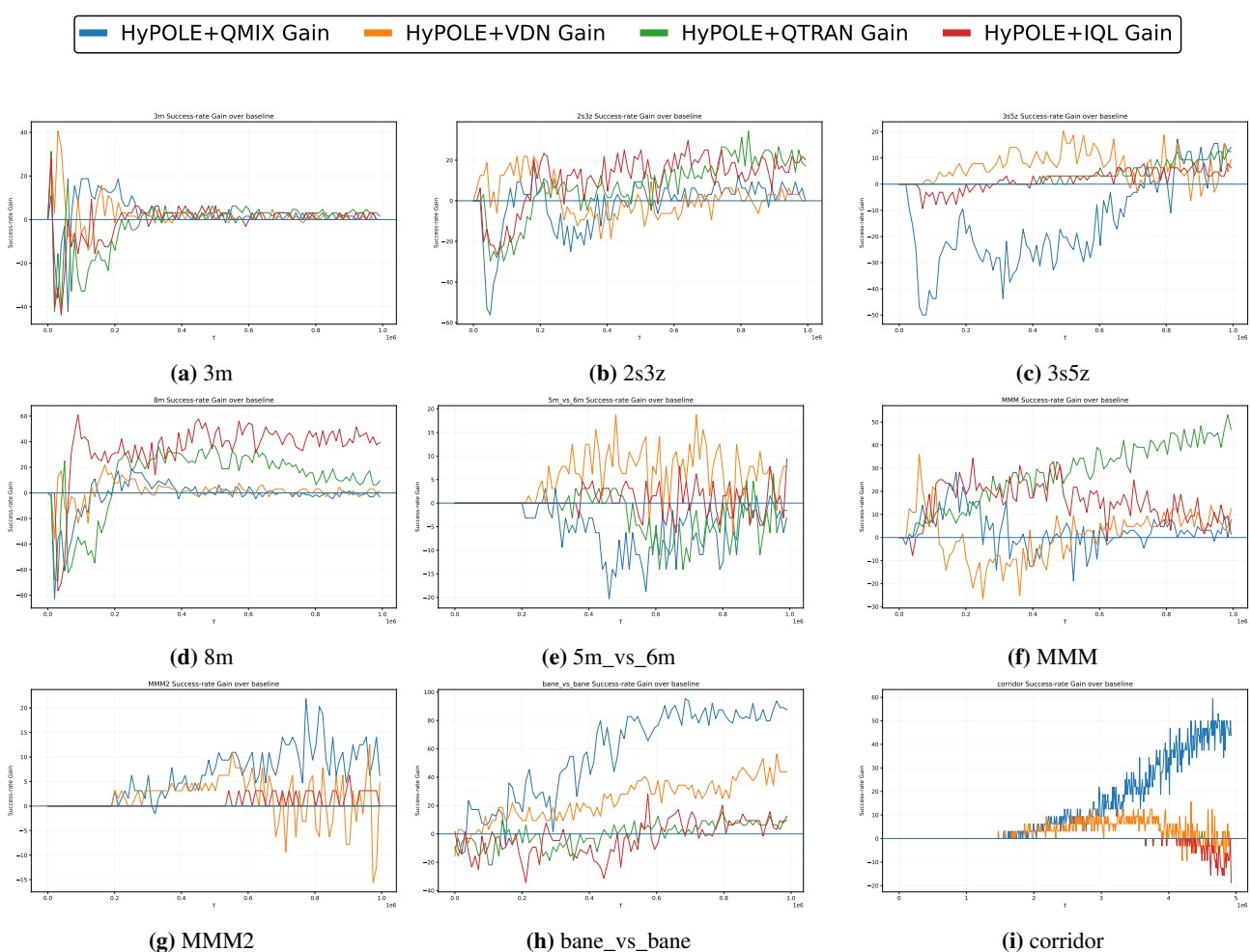

**Figure 10.** Median Gain in Percentage of HYPOLE compared to their respective baselines in SMAC.

better performance than HYPOLE +QTRAN (brown line). On the `8P_3F2V` map (Figs. 12g to 12i), HYPOLE +MARL outperforms the vanilla baselines in all cases; notably, HYPOLE +VDN (red line) and HYPOLE +IQL (gray line) perform substantially better than the vanilla variants across all baseline reward functions. On the `10P_3F2V` map (Figs. 12j to 12l), HYPOLE +MARL outperforms the vanilla baselines in all cases; in particular, HYPOLE +VDN (red line) and HYPOLE +QMIX (orange line) significantly improve over the vanilla variants under all baseline reward functions.

### H.2. ∀∃ vs. ∀∀ Specifications.

Since `MMM2` includes a Medivac unit, we evaluate the $\varphi_{\text{medivac}}$ specification using ∀∃-HyperLTL formulas. As shown in Fig. 15a, HYPOLE (∀∃)+QMIX (orange line) performs slightly better than HYPOLE (∀∀)+QMIX (blue line). We also provide same analysis for 10P_3F2V map in WildFire Fig. 15b and we saw HYPOLE (∀∃)+QTRAN (brown line) performs better than HYPOLE (∀∀)+QTRAN (purple line).

### H.3. Visualization of tactics Learn by HYPOLE on SMAC

Fig. 20 shows snapshots after 1M training steps in SMAC. In Fig. 20a using $\varphi_{\text{defense}}$, the agents instead prioritize survival and retreat, moving away from enemies rather than committing to an engagement. Under $\varphi_{\text{medivac}}$ Fig. 20b, the Medivac consistently moves toward the ally with the lowest health bar, indicating that the learned policy captures the intended support behavior. At the same time, the remaining agents exhibit focused firing by concentrating fire on a specific enemy target.

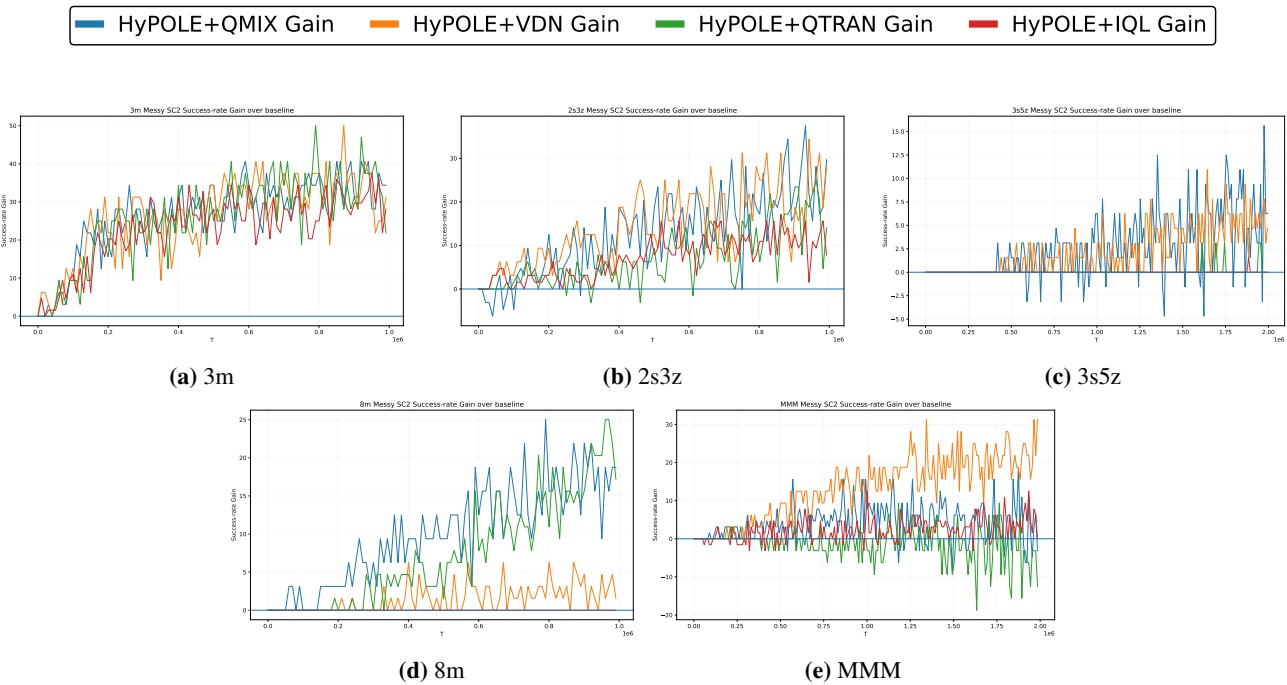

**Figure 11.** Median Gain in Percentage of HʏPOLE compared to their respective baselines in MessySMAC.

**Table 3.** Mean execution time on the MMM map comparing MARL with and without HʏPOLE. Results for the ∀∃ formula are averaged only over VDN, QMIX, and QTRAN.

| Method | 200K Steps | 400K Steps | 600K Steps | 800K Steps | 1000K Steps |
|---|---|---|---|---|---|
| **MARL** | 2202 | 4294 | 6305 | 8377 | 10385 |
| HʏPOLE (∀∀) + **MARL** | 2211 | 4453 | 6637 | 8918 | 11100 |
| HʏPOLE (∀∃) + **CTDE** | 3366 | 6821 | 10260 | 13659 | 16871 |

Both Figs. 20a and 20b generated by VDN.

### H.4. HʏPOLE Computational Overhead

Next, we evaluate the computational overhead introduced by HʏPOLE within the CTDE training pipeline. We run an experiment on MMM, with results reported in Table 3. In the ∀∀ setting, HʏPOLE incurs a negligible overhead of about 6% at 1000K environment steps. In contrast, in the ∀∃ setting, the overhead increases to 62%, which is substantial. This experiment is run on an AMD EPYC 7742 64-core CPU and a single NVIDIA RTX A6000 GPU.

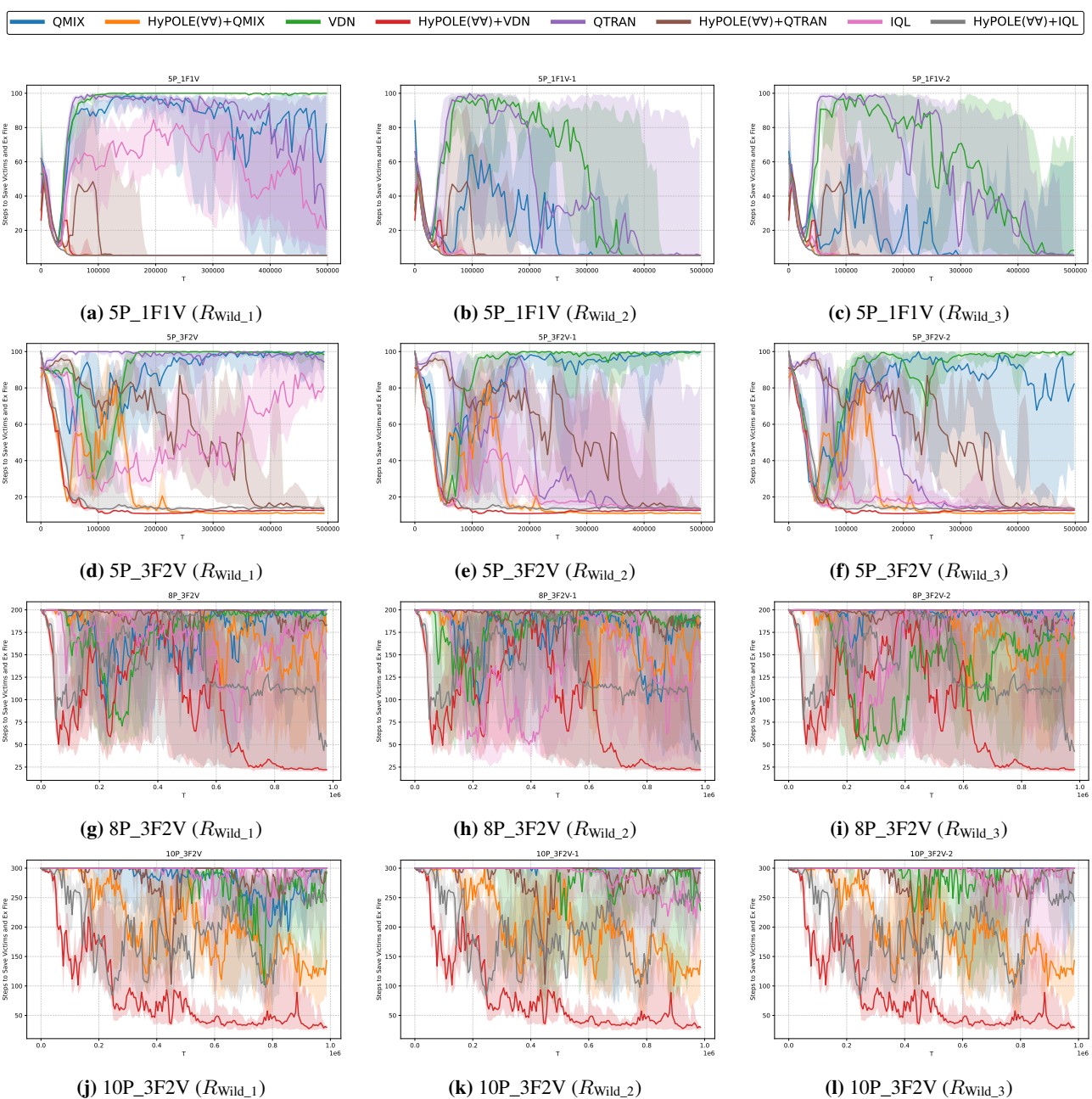

**Figure 12.** Learning curves on different WildFire Maps, showing median Steps to Extinguish Fire and Save Victim with 25-75% percentile.

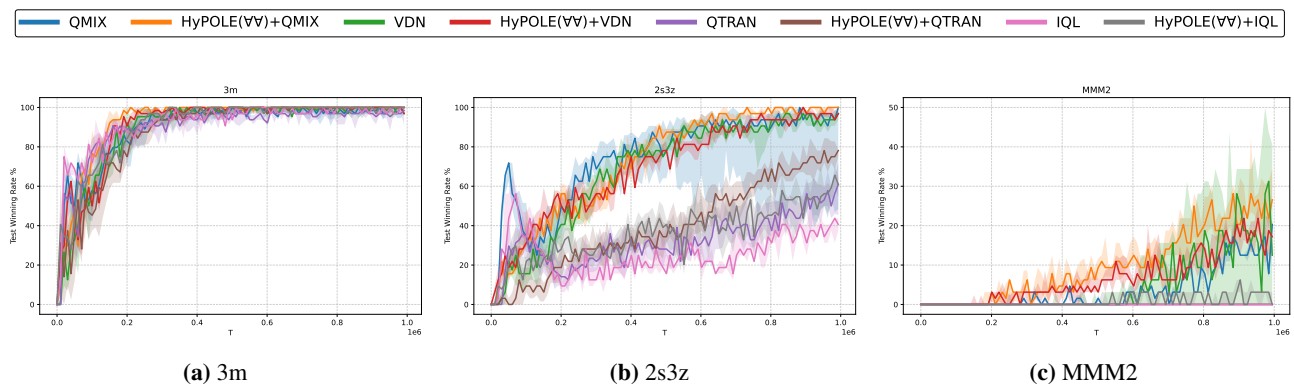

**(a)** 3m  **(b)** 2s3z  **(c)** MMM2

**Figure 13.** Learning curves on remaining SMAC maps, showing median test win rate with 25-75% percentile.

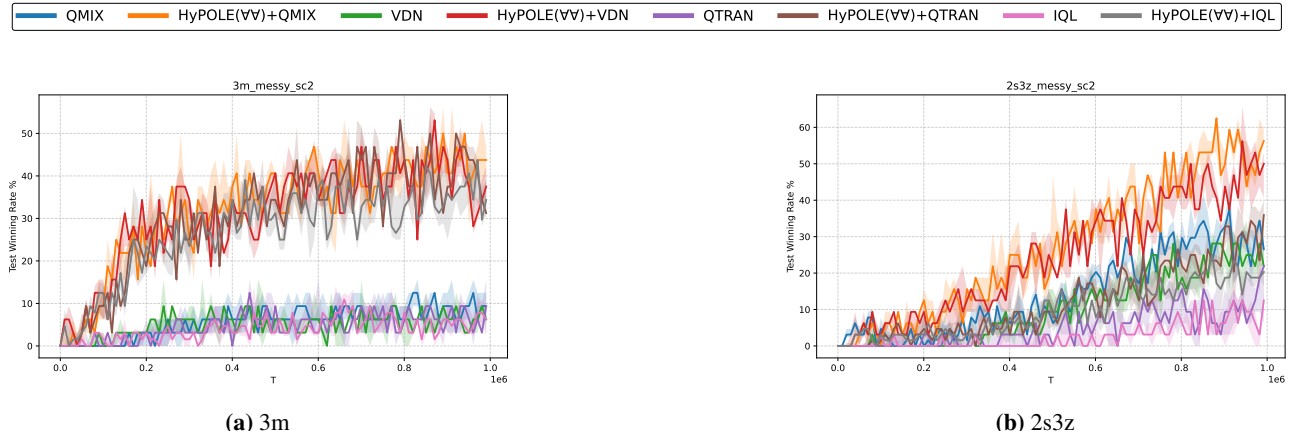

**(a)** 3m  **(b)** 2s3z

**Figure 14.** Learning curves on remaining MessySMAC maps, showing median test win rate with 25-75% percentile.

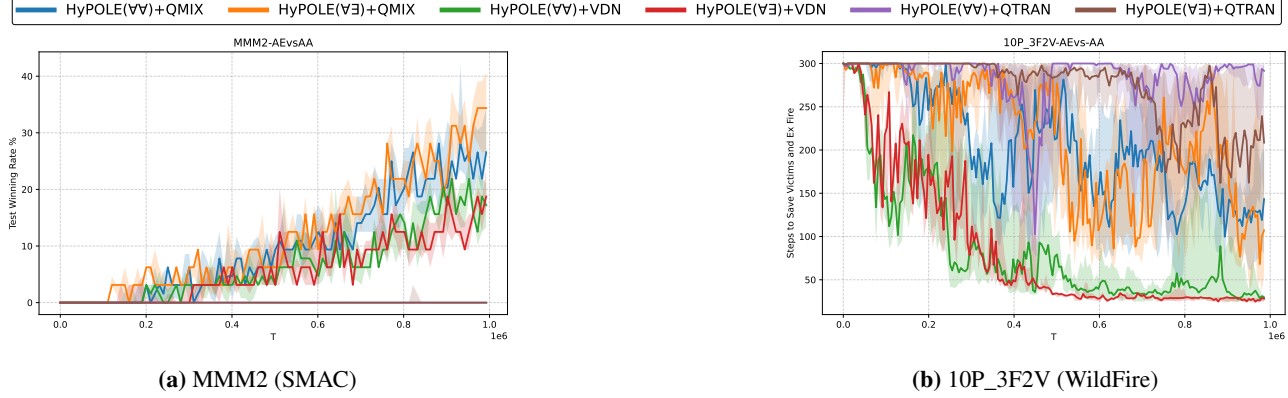

**(a)** MMM2 (SMAC)  **(b)** 10P_3F2V (WildFire)

**Figure 15.** ∀∀ vs. ∀∃ learning curves across SMAC, and WildFire, showing median test win rate with 25–75% percentiles.

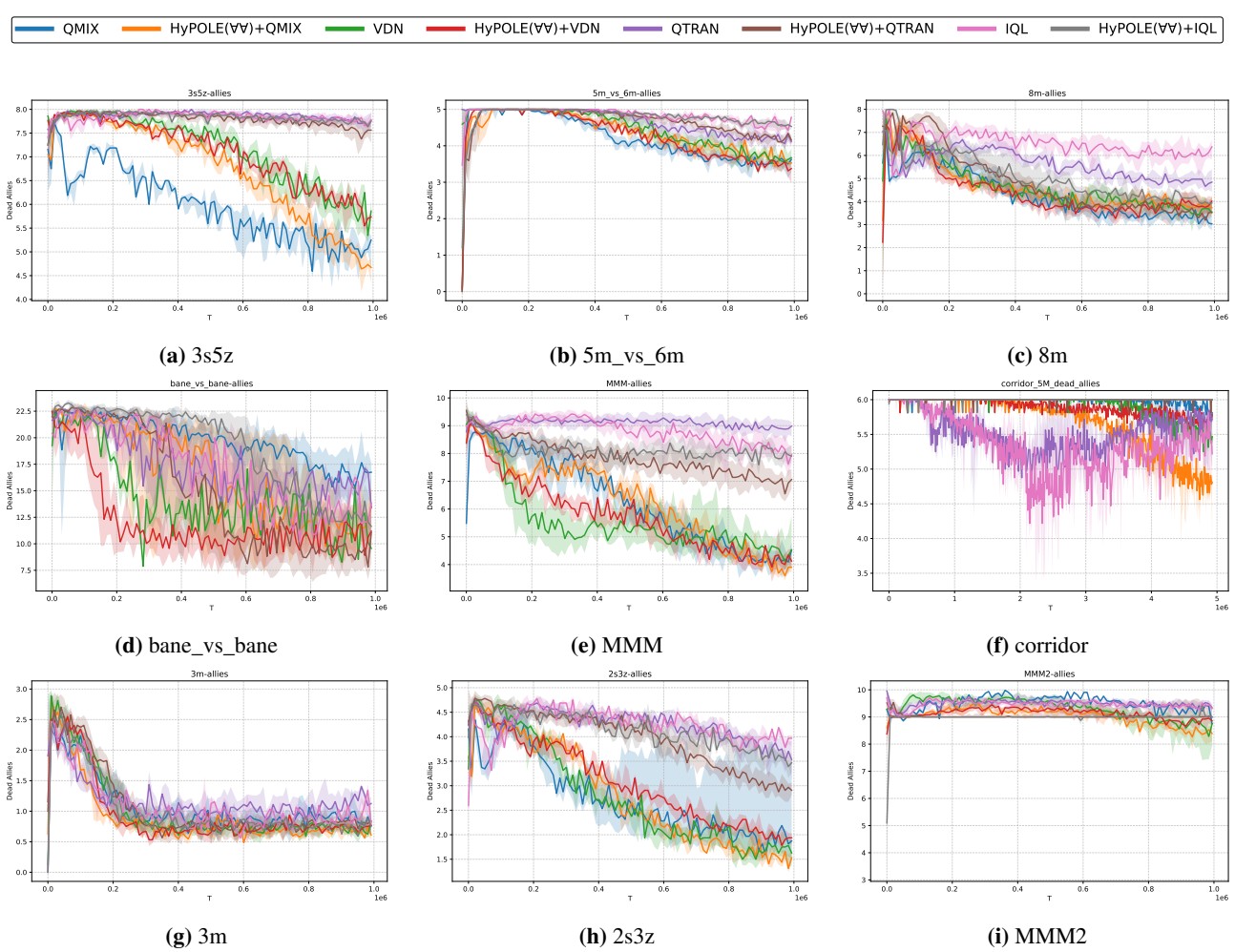

**Figure 16.** Learning curves on different SMAC combat maps, showing median dead ally agents with 25-75% percentile.

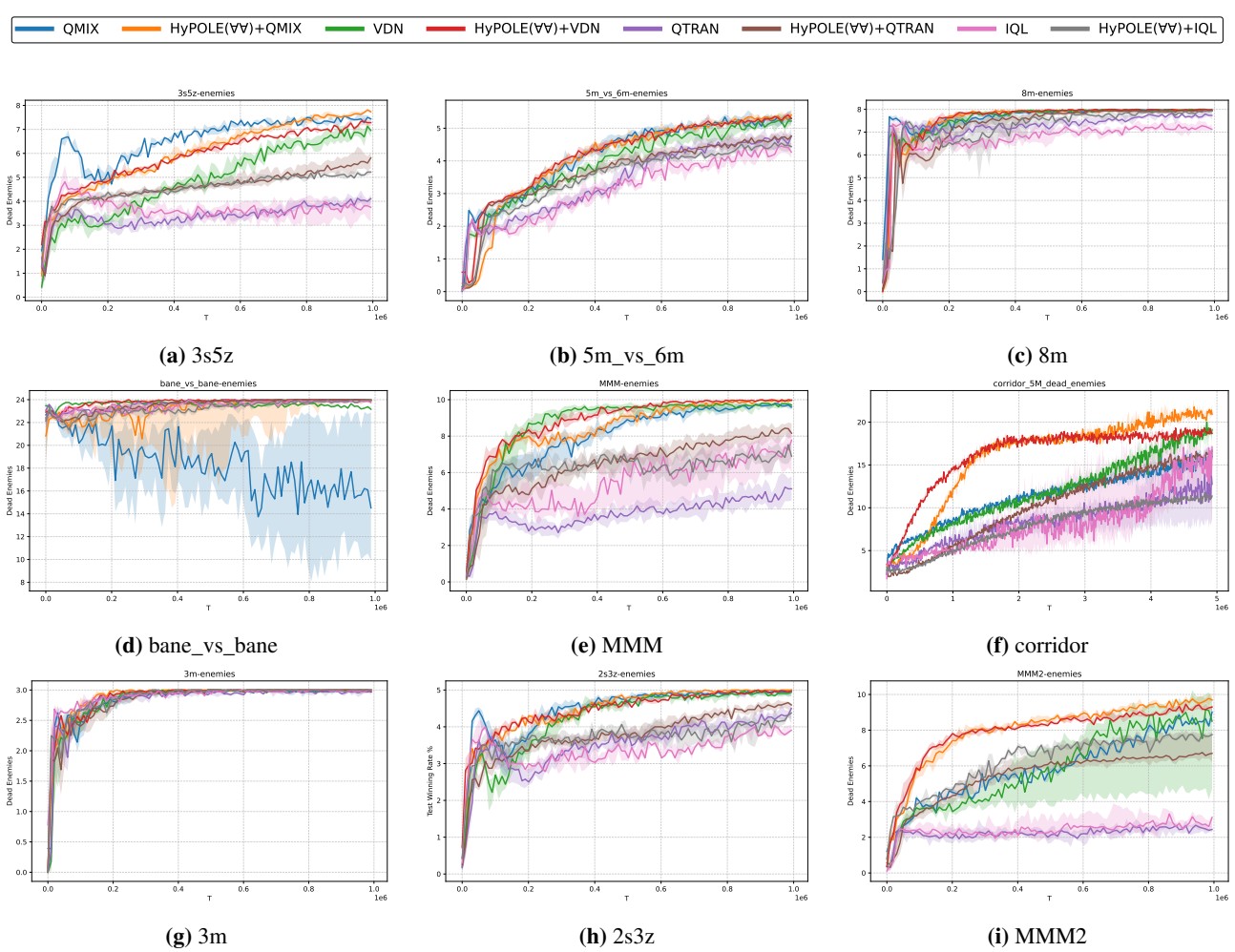

**Figure 17.** Learning curves on different SMAC combat maps, showing median dead enemy agents with 25-75% percentile.

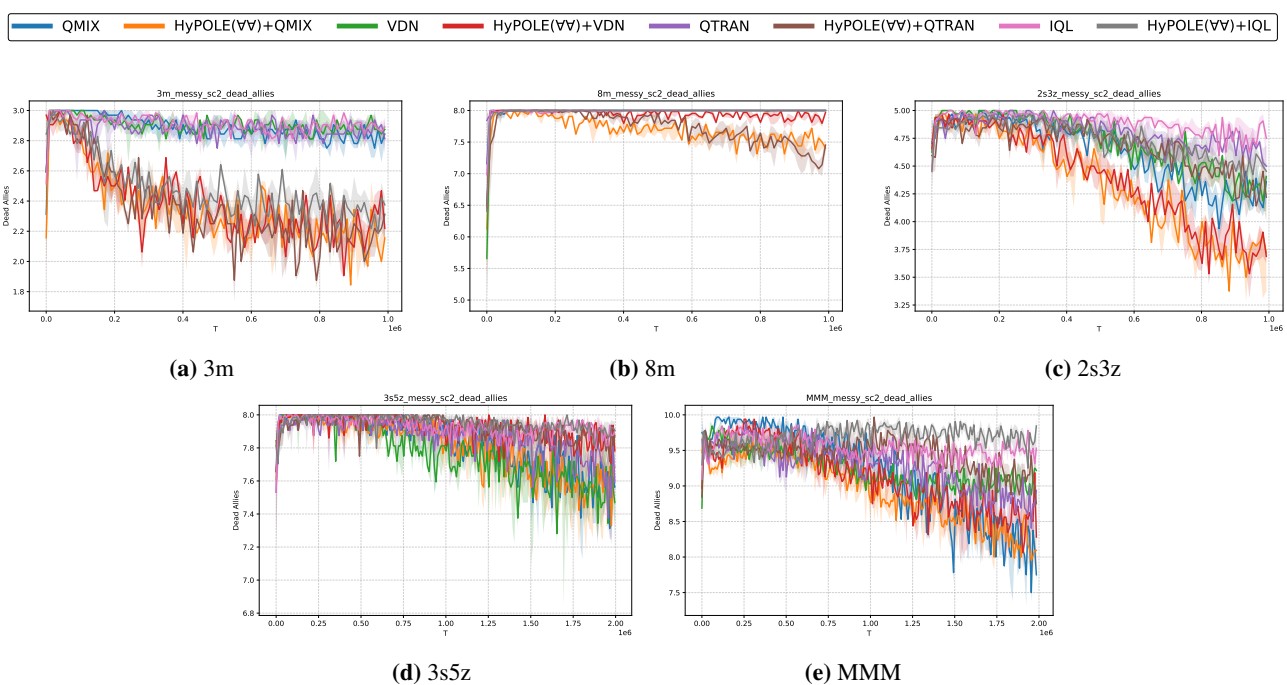

**Figure 18.** Learning curves on different MessySMAC combat maps, showing median dead ally agents with 25-75% percentile.

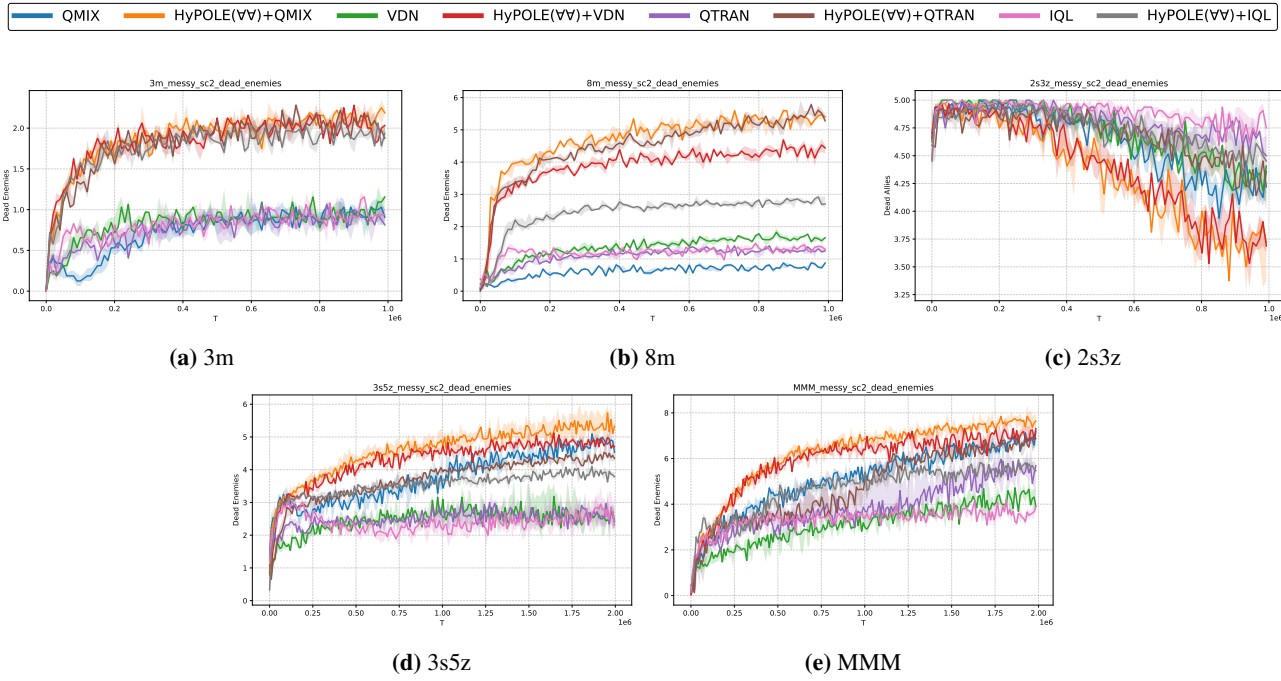

**Figure 19.** Learning curves on different MessySMAC showing median dead enemy agents with 25-75% percentile.

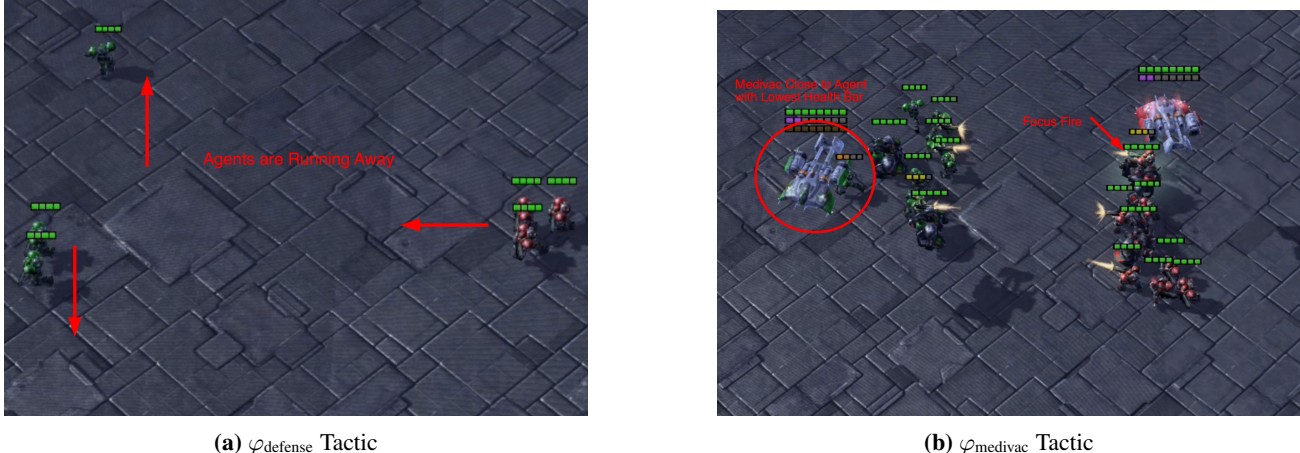

**(a)** $\varphi_{\text{defense}}$ Tactic

**(b)** $\varphi_{\text{medivac}}$ Tactic

**Figure 20.** Snapshots of scenarios under HyperLTL-specified tactics.

