# OpenReview forum: "HyPOLE: Hyperproperty-Guided Multi-Agent Reinforcement Learning under Partial Observation"
_ICML.cc/2026/Conference — ICML 2026 regular_

### Official Review · Reviewer_4TgS · 2026-02-13

**Soundness:** 1
**Presentation:** 3
**Significance:** 1
**Originality:** 1
**Overall Recommendation:** 1
**Confidence:** 5

**Summary:**

This paper propose HyPOLE, a method that uses hyperproperty specification to guide multi-agent reinforcement learning in partially observable environments. HyPOLE first Skolemize the traces of trajectories and then generate robust score to guide the learning of Q-value. HYPOLE is implemented on top of the SMAC and wilefire environments with several ablation studies.

The main contribution of this paper is leveraging the concept of Skolemization, which I find very interesting and may have great potential in the ML community.

**Compliance With Llm Reviewing Policy:**

Affirmed.

**Final Justification:**

**I will lower my score to 1 (strong reject).**

The primary reason is that, throughout multiple rounds of rebuttal, the authors have consistently failed to directly address my core concerns. Besides, based on the discussion, **some reviewers either increased their scores with low confidence** or **failed with carefully checking important related work [1]** in their assessment.

I want to emphasize that **I respect the effort behind every submission.** I understand that research work often takes months or even years. However, after carefully reading the paper, and rebuttal, I am convinced that this work should not be accepted. Therefore, **I provide a detailed summary of my concerns below to raise AC’s attention.**

**Key Questions For Authors:**

# Questions

**Q1.**

As W4 mentioned, this method relies on Q-learning based method and only be tested in discrete-action scenarios.  Could Hypole be extended to the continuous-action algorithms such as MADDPG or MATD3?

**Q2.**

Since robustness scores are derived from manually specified formulas/rules, do these  formulas/rules restrict exploration? For example, in the corridor, agents can easily learn to gather at the choke points; meanwhile, they lose the chanses to explore other strategies. Therefore, if the baselines are fine-tuned and are allowed to explore extensively, they theoretically can converge to better policies since their reward signals are more expressive than the  formulas.

**Q3.**

In the section of 5.2, the traces are formulated over finite-time horizon (0:k). However, in the most of the RL literature, inifite-time horizon is typically assumed to make sure the value function (bellman equation) can converge. Therefore, I am curious how the Q function is defined in this paper. If the Q function is defined over 0 to k, why is it not explicitly time-dependent, and how is convergence ensured?

**Limitations:**

I haven't found discussion of limitations of this paper. Please refer to the Weaknesses section for detailed suggestions .

**Strengths And Weaknesses:**

# Strengths

1.The idea of Skolemization is novel to the MARL community.
2. The paper is well-structured and generally clear.

# Weaknesses

**W1. Motivation**

The motivation of leveraging specification is **virtually all existing approaches attempt to search for policies for objectives that are of the form $\forall\forall^*$, while many multi-agent requirements are of the form $\forall \exists$.** However, I believe this claim mainly applies to cooperative MARL,  especially since the scope of this paper heavily relied on the assumption of IGM. The authors should provide more citations or evidence to justify this claim.

**W2. Limtied Novelty**

This paper highly resembles the previous work [1]. The [1] also gives the formulas first, then Skolemize, then use robustness score to replace reward to guide the learning of Q function. The main difference I can tell is that in this paper, the robustness score is to guide the learning of Qtot instead of individual Q value. Though cooperative MARL is different from single agent RL, the MARL insights are very limited.

**W3. Usage of formulas vs revised reward signal**

I notice the HyPOLE relies on the manually defined formulas in Appendix E.1, which have some prior knowledge or insights that can guide agents to learn good policies like focus-fire, defending in corridor.  But in the original reward settings, the agents need to explore many runs to find this kind of ***good policies***. So it is difficult to disentangle where the gain of HyPOLE comes from. For example, the author claimed HyPOLE is pretty effective in Corridor scenario, but all the baselines use the original reward signal defined in SMAC itself, while the HyPOLE's formula has some additional insights like ***defending in choke point***. It is unclear whether the gains come from specification-related insights or from the learning framework itself (e.g., Skolemization and robustness score). If that gain comes from specification, why don't we just revise the reward signal directly?

**W4. Limited Experimental Settings**

I think that's the one of the main limitations of this paper. Firstly, the wildfire environment appears as a didactic example, since it restricted within discrete state and discrete action space. And the SMAC is also a discrete-action benchmark. Therefore, the experimental results only demonstrate HyPOLE can be deployed on the discrete action envrionments. Besides, the performance gain of HyPOLE is trivial. Based on the Figure 4, the QMIX performs comaprable in most of the settings. In (b) and (d), vanilla QMIX even appears to  converge to  better policies.

**W5. Minor Comments**

1. Some acronyms should be defined first before use in the abstract like MARL and CTDE.

2. Partially observable Markov decision process (POMDP) appeared twice. One in the third paragraph of Introduction, while another appeared in 3.1.


[1]Hsu, Tzu-Han, Arshia Rafieioskouei, and Borzoo Bonakdarpour. "HypRL: Reinforcement Learning of Control Policies for Hyperproperties." Neurips 2025.

---

> ### Author Rebuttal · Authors · 2026-03-30
>
> We thank the reviewer for the detailed review. In our response, please refer to [Link](https://tinyurl.com/28cb2hee).
>
> >Q1&W4: MADDPG and MATD3? Limited Experiments.
>
> HyPOLE can be extended to actor-critic (AC) methods (e.g. MADDPG, MATD3), but at the **cost of soundness.** We use value-based CTDE to optimize a centralized objective and derive decentralized optimal policies via IGM to guarantee Thm.5.3., which AC methods cannot do. We will clarify this in the final version.
> For rebuttal, we **add evaluations on MessySMAC** [Phan et al. ICML'23], which introduces observation stochasticity and randomized initialization: In 3m(`Link:Fig.2a`), HyPOLE reaches up to 50% win rate. In 8m(`Link:Fig.2b`), the baselines have no success, but HyPOLE+(QMIX/QTRAN) reach up to 25% win rate. This shows that HyPOLE remains effective under randomized initialization and observation stochasticity.
>
> >Q2&W3: Isn't using HyperLTL restrict exploration?
>
> HyPOLE is a spec-guided MARL framework whose key advantage is leveraging HyperLTL to guide learning, rather than exploring all policies. It enables efficient discovery of desired behaviors, especially those with **inter-agent dependencies and temporal behaviors**. Some formulas (e.g., maintaining health levels) act as simple guiding constraints. Your comment also aligns with our ongoing direction of automatically mining HyperLTL from natural language and building an end-to-end pipeline from NL to specs to policies.
>
> >Q3: How is convergence ensured?
>
> In Sec.5.2, traces are finite as we transition from a satisfaction problem to an optimization problem, while being consistent with **finite-trace HyperLTL semantics**. However, the Q-function in 276-77 (left) is **defined recursively and updated iteratively** from step k to k+1, so it is not restricted to finite-horizon. We will clarify it in the final version.
>
> >W1: Lack of support for motivation.
>
> Prior spec-guided MARL work uses non-hyper logics with single-trajectory objectives $\forall\forall$. However, many MARL problems are relational, where $\forall\forall$ is too strong and $\forall\exists$ is more natural. For example, in HypRL [1]: Although single-agent, PCP and DeepSea are reformulated as $\forall\exists$ specs; SafeRL (collision avoidance) and contingency planning are also naturally $\forall\exists$. Beyond HypRL, [Beutner et al. ICAPS'24] also includes such cases. To the best of our knowledge, HyPOLE is the first to study $\forall\exists$ specs in partially observable MARL, so the related works remain limited. Nevertheless, [Yu Wang et al. ICRA'20] and [Hsu et al. TACAS'21] both support that MA planning problems are naturally expressible in HyperLTL.
> We agree that HyPOLE targets cooperative MARL, but it opens the door to richer logics such as HyperSL [Beutner et al. AAMAS'24], which supports both cooperative and adversarial settings. We will revise the motivation section.
>
> >W2: Limited Novelty compared to HypRL.
>
> We respectfully disagree. HyPOLE provides a **strictly more general framework** than HypRL. `Link:Fig.1` shows that Dec-POMDPs subsume MDPs, meaning that HyPOLE's algorithm can also be applied to the problems in HypRL, so it is not just replacing individual Q-values with $Q_{tot}$. Below are all non-trivial contributions compared to HypRL:
> 1. Dec-POMDP lifting (Sec.5.3): Instead of centralized MDP, where a single centralized Q-function is learned, HyPOLE lifts the problem to Dec-POMDPs and proves that CTDE yields optimal decentralized policies.
> 2. Quantifier-sensitive pipelines (Fig.1): HyPOLE explicitly distinguishes $\forall\exists$ vs. $\forall\forall$ through different training pipelines, yielding qualitatively different behaviors (Fig.6).
> 3. Observation history-based Skolemization: HyPOLE requires Skolem functions over histories (not traces), complicating both sampling and optimization (training the encoder-decoder to predict the input of Skolem function).
>
> >W3: Why not shape reward directly?
>
> A HyperLTL formula includes: The quantifier structure, via Skolemization, captures inter-agent dependencies, while the inner LTL body, via robustness, captures temporal behaviors. These are not expressible through standard scalar reward design, so the gains do not come from simply revising the reward.
>
> >W4: HyPOLE+QMIX performs comparably to QMIX.
>
> We respectfully disagree. For rebuttal, we extended corridor to 5M steps in `Link:Fig.4`, where QMIX+HyPOLE reaches up to 60% win rate while QMIX remains negligible. Also, `Link:Fig.5` shows the gains achieved by HyPOLE across all maps. QMIX+HyPOLE outperforms the baseline on almost all maps except 5m_vs_6m and 3s5z, and even in 3s5z it surpasses the baseline in the final 200k steps. In harder scenarios (MMM2, bane_vs_bane, and corridor), QMIX+HyPOLE shows **substantial improvements: up to 90% gain in success rate** on bane_vs_bane.
>
> We will address all minor comments.
>
> **If our responses have addressed your comments, we respectfully ask you to consider raising the score.**

---

> > ### Author Rebuttal · Reviewer_4TgS · 2026-04-01
> >
> > My main concerns remain largely unaddressed. Please check my new reply for details.
> >
> > **Q1 & W4**
> >
> > My key question regarding whether the method can be extended to continuous action spaces was not addressed. I expected supporting results in continuous control environments (e.g., MPE or MuJoCo). Instead, the rebuttal provides additional SMAC experiments, which do not resolve this limitation.
> >
> > **W2**
> >
> > The technical novelty remains unclear. From my understanding, the main difference is replacing individual Q-values with a joint $Q_{\text{tot}}$, which is already standard in CTDE-based MARL. While the authors claim a Dec-POMDP formulation, this does not appear to introduce new technical challenges or methodological contributions.
> >
> > The use of $\forall\forall$ and $\forall\exists$ specifications is also not fully justified. In particular, QMIX/SMAC operate under cooperative settings that rely on the Individual-Global-Max (IGM) assumption. It is therefore unclear how $\forall\exists$ (AE) semantics, which introduce inter-agent dependency, are compatible with this setting. Such distinctions would be more convincing in competitive or mixed settings rather than purely cooperative ones.
> >
> > **W3 & W4**
> >
> > I am not convinced by the justification of using HyperLTL instead of reward shaping. If the specifications can be constructed in the form of formulas, it remains unclear why equivalent shaped rewards cannot be designed. The rebuttal does not provide a clear distinction between these two approaches.
> >
> > Furthermore, the source of the performance gain is unclear. It is not evident whether the improvements come from the proposed framework (e.g., Skolemization) or simply from incorporating predefined domain knowledge through the specifications.
> >
> > Overall, the rebuttal does not resolve the core concerns regarding applicability, novelty, and conceptual clarity. Therefore, I will not increase my score.

---

> > > ### Author Response · Authors · 2026-04-05
> > >
> > > Q1&W4
> > >
> > > We understand your question about continuous-action. The setting of HyPOLE is on discrete action and extending it to continuous actions isn't a trivial task and certainly can't be done within a week or two. While we fully appreciate the importance of continuous settings, it is simply not the focus of this submission.
> > >
> > > W2
> > >
> > > We would like to point you to a factual error: we are not replacing individual Q-val with $Q_{tot}$. First, we note that[1] focuses on a centralized policy learned using a single joint Q, rather than individual Q. Second, our problem formulation is fundamentally different from[1]. This shift introduces several new challenges, including handling Skolem func without access to other agents’ traces due to decentralization, working only with obs histories, and constructing a sound pipeline that guarantees our solution in partially obs, decentralized setting. Please note [ICML](https://tinyurl.com/mv5rf538) doesn't require an entirely new method for a contribution to be considered novel.
> > > It seems the reviewer's impression is that a $\forall\exists$ spec is only applicable to adversarial settings. This is not true. 2 cooperative agents can still satisfy a $\forall\exists$ spec as demonstrated in WildFire in Sec.3. In fact, this has nothing to do with how QMIX/SMAC works. As another example, we also refer you to 2 case studies in[1], where two single-agent problems (PCP&DST) are specified by $\forall\exists$ spec.
> > >
> > > W3&W4
> > >
> > > Traditional reward shaping have 3 shortcomings:(1) lack of mathematical rigor in stating objectives and constraints (2)inability to capture temporal behavior of agents, and (3)relational and interdependence of agents tasks. The first two has already been established in the literature by showing the value of formal spec-guided RL, mainly focused on LTL as the in single-agent settings (even LTL hasn't been studied with CTDE in spec-guided MARL, and HyPOLE is the first work). We invite the reviewer to study [Link](https://tinyurl.com/j82pejms) for a detailed discussion in this area. In addition to the first two issues, our work significantly expands spec-guided MARL with respect to the third issue as well, which turns out to be challenging as well as rewarding. Our technique enables $\forall\exists$ spec, which not only widens expressiveness, but also significantly improves the performance of MARL.
> > > We also clarify that even designing shaped rewards requires domain knowledge about the env and the task, so the need for such knowledge isn't unique to our work.
> > >
> > > ---
> > > # Addressing Reviewer 4TgS Final Justification
> > > We would like to provide additional clarification on the novelty of our work compared to[1], as well as on the challenges in continuous action-space settings. We hope this further clarifies and addresses the reviewer’s concerns.
> > > # Comparison to HypRL[1]
> > > We respectfully clarify that HyPOLE is fundamentally different from [1]. In particular, HyPOLE addresses **decentralized policy synthesis under partial observability**, whereas [1] focuses on **centralized policies in fully observable settings**. Thus, HyPOLE is already novel at the level of problem formulation. To make this distinction clearer, we present the following **comparison table**:
> > > ||HyPOLE|HypRL[1]
> > > |-|-|-
> > > |**Partial Obs**|✅|❌
> > > |**Decentralize Policies**|✅|❌
> > > |**Skolemization**|**Complex**:decentralized & partial obs (no access to $\forall$-agent traces).|**Simple**:centralized & full obs (access to $\forall$-agent traces).|
> > > |**$\forall\forall$vs.$\forall\exists$**|**Different pipelines**:Two forms are handled differently, with experiments showing their behavioral distinction.|**Same pipeline**:Two forms are handled the same, no behavioral distinction.
> > > |**Ablation on Formulas**|✅|❌
> > > |**Robustness**|**New formulations**:Handles **partial obs** (e.g., Eqs.2&3) with a new proof techniques for Thm.5.1.|**Original formulations**: Limited to **fully obs setting** only.
> > > |**Scalability**|Upto **24 agents**, including challenging maps (e.g., corridor).|Only upto **3 agents** on small grid worlds.
> > > |**Formulas**|**Complex formulas**: Nested temporal operators and a comparison of different strategies (see App.E.1).|**Straightforward formulas**: No comparison of different strategies (see Tab.1 in [1]).
> > > ## Overall, HyPOLE's is a non-trivial generalization of HypRL.
> > > # Continuous action-spaces
> > > There are two main families in CTDE: value-based(VB) and actor-critic(AC) methods. AC methods can handle continuous action, unlike VB methods, but they do not provide an IGM-like property, which is essential for our theoretical result in Thm.5.3 and **soundness of our work**. Therefore, under our current theory, HyPOLE cannot directly operate in continuous-action settings, although continuous state spaces are not a problem. Also, in multi-agent MuJoCo [Schroeder de Witt et al., 2020], Tab. 1 shows that the goals are mostly to make the `robot move as fast as possible`, which lacks the temporal and relational structure targeted by HyperLTL.

---

### Official Review · Reviewer_T79a · 2026-02-28

**Soundness:** 3
**Presentation:** 3
**Significance:** 3
**Originality:** 3
**Overall Recommendation:** 4
**Confidence:** 3

**Summary:**

The paper proposes a formal approach towards MARL under constraints and partial observability, introducing the HyPOLE framework, which uses the HyperLTL logic to specify objectives and constraints in a MARL setting to enable more expressive specifications than ad hoc reward shaping methods. The specification is then used to derive the (Dec-)POMDP and rewards, which can be solved via off-the-shelf MARL methods like VDN, QMIX, and QTRAN. HyPOLE is evaluated in the old StarCraft II benchmark, showing similar performance to their non-enhanced baselines.

**Compliance With Llm Reviewing Policy:**

Affirmed.

**Final Justification:**

The rebuttal addressed my major concerns. The additional results strengthen the significance of the work and make the paper acceptable in my view.

**Key Questions For Authors:**

None

**Limitations:**

Limitations and potential negative societal impact have not been discussed. The proposed method appears to introduce additional overhead, i.e., specification effort and Step 1 (HyperLTL Skolemization) and Step 2 (Learning with Quantitative Semantics), on top of the MARL machinery.

**Strengths And Weaknesses:**

**Originality**

The technical contribution is intriguing, as the paper combines symbolic techniques with neural MARL methods in a principled way. I consider originality the biggest strength of this paper.

**Soundness**

The paper appears to be technically sound. I have not checked the details, though (see below).

**Presentation**

The paper is generally well-written. It took some time to digest Sections 3.2 and 4 due to their notation-heavy texts (there is a chance that I missed crucial details while parsing through these sections).

Although the abstract promises an evaluation in a Wildfire benchmark, the main paper only presents StarCraft II results.

**Significance**

Theoretically, the contribution could be significance. However, empirically, I am missing convincing evidence to confirm the signifcance:
- In most scenarios, the non-HyPOLE counterpart performs very similar (or better). For a tractable proof-of-concept, I suggest an evaluation in a small-scale domain, where it can be shown theoretically and practically that HyPOLE outperforms its baselines.
- I do not see, where constraints play an important role in the presented domains. I suggest to evaluate HyPOLE on safety-relevant domains, where conventional MARL techniques are expected to fail [1]
- StarCraft II is an outdated benchmark due to its deterministic nature (fixed initial states and deterministic observations, which enable open-loop control - without any observations) [2,3,4]. I suggest to evaluate in SMACv2, an updated version of the StarCraft II benchmark with randomized initial states [2].

**Literature**

[1] Gu et al., "Safe Multi-Agent Reinforcement Learning for Multi-Robot Control", AIJ 2023

[2] Ellis et al, "SMACv2: An Improved Benchmark for Cooperative Multi-Agent Reinforcement Learning", NeurIPS Benchmarks 2023

[3] Lyu et al., "A Deeper Understanding of State-Based Critics in Multi-Agent Reinforcement Learning", AAAI 2022

[4] Phan et al., "Attention-Based Recurrence for Multi-Agent Reinforcement Learning under Stochastic Partial Observability", ICML 2023

---

> ### Author Rebuttal · Authors · 2026-03-30
>
> We thank reviewer T79a for their detailed review. Below, we address each comment. Please see [Link](https://tinyurl.com/28cb2hee) where we cite `Link` in the rebuttal.
>
> >It took time to digest Sec3.2...texts.
>
> Sec.3.2 introduces the syntax and semantics of HyperLTL and Sec.4 formalizes the problem setting for mathematical rigor. If accepted, we will add more explanations to improve readability.
>
> >Although...StarCraft results.
>
> Due to space, the wildfire results are placed in the appendix. If accepted, we will use the additional page to bring these results into the main paper.
>
> >In most scenarios, the non-HyPOLE...or better.
>
> This is a factual error. On average, in 7.5 out of 9 cases, HyPOLE improves over the vanilla baselines, and some of these gains are substantial. To demonstrate this, Fig.5 in `Link` shows the gains across all maps. QMIX+HyPOLE(blue) outperforms QMIX on almost all maps except 5m_vs_6m and 3s5z, where in 3s5z it still surpasses in the final steps. In hard scenarios such as MMM2 (Fig.5g), bane_vs_bane (Fig.5h), and corridor (Fig.5i) (extended to 5M steps, for rebuttal, the corresponding plots in Fig.4 in `Link`) QMIX+HyPOLE show **significant improvements**, with **up to 90% gain in success rate** on bane_vs_bane. Similarly, VDN+HyPOLE(orange) improves performance in most cases, except 2s3z where the behavior is more unstable, while achieving **substantial gains** on bane_vs_bane and challenging map 5m_vs_6m. Finally, QTRAN+HyPOLE(green) outperforms the baseline in nearly all scenarios except 5m_vs_6m, and achieves **notable improvements** on MMM with **upto 50% gain**(Fig.5f). Furthermore, IQL+HyPOLE(red) achieves better results on almost all maps, specifically about **upto 60% gain** on 8m(Fig.5d).
>
> >For a tractable...its baselines.
>
> We adapted and extended the Wildfire Scenario [HypRL Hsu et al., 2025] to partially observable setting. This environment illustrates why quantifiers play a key **theoretical** role in HyPOLE. As shown in our running example (lines 144-164 right), changing $\forall\exists$ to $\forall\forall$ reduces the probability of satisfaction from 1 to 0.75 since $\forall\forall$ is a stronger requirement. To further demonstrate the **practical impact** of this distinction, for the rebuttal we also conducted additional experiments on two wildfire maps (Fig.3 in `Link`). On both maps, VDN+HyPOLE($\forall\exists$)(purple) converges faster than VDN+HyPOLE($\forall\forall$)(orange), and on the 8P_3F2V map this improvement is substantial. We also observe gains for QTRAN+HyPOLE($\forall\exists$)(brown) over QTRAN+HyPOLE($\forall\forall$)(green) on both maps, especially on 8P_3F2V.
>
> >I do not see...expected to fail[1].
>
> First, we clarify that HyPOLE is **spec-guided MARL**, not **constrained RL** such as shielding. HyPOLE encodes safety constraints in HyperLTL formula, and expresses richer strategies and use them to guide MARL. In both SMAC and wildfire, we already consider such constraints. For example, in SMAC, our defensive spec aims to avoid allied casualties, as shown in Fig.5b. We also show in Fig.5a that different HyperLTL formulas, such as kiting and focus fire, can largely affect performance even under the same overall objective. Regarding [1], our framework is built on value-based CTDE to preserve **soundness**, since these methods provide an IGM property required by our theoreom 5.3, whereas actor-critic methods do not. As a result, our current framework does not handle **continuous action spaces**, so we cannot evaluate on continuous-control domains such as MARobosuite (the only partial obs case in [1]).
>
>
> >StarCraft is an outdated...initial states[2].
>
> Thank you for the suggestions. We identified MessySMAC [4], which introduces observation $\phi$ and randomized initialization (K random steps before each episode starts). We evaluated HyPOLE with CTDE on 3m and 8m and (Fig.2 in `Link`). In 3m with $K=10,\phi=.15$ Fig.2a, HyPOLE reaches up to **50% win rate**. In 8m with $K=12,\phi=.15$, baselines achieve no success, while HyPOLE+(QMIX/QTRAN) reach up to **25% win rate**. This shows HyPOLE remains effective under observation and initialization stochasticity. We will add these maps in the final version. Regarding [3], we did not find their code. Regarding SMACv2, we believe MessySMAC already captures randomized initialization, while additionally introducing observation stochasticity. Therefore, we believe [4] is a better benchmark to address the reviewer’s concern, but we welcome your perspective.
>
> >Limitation
>
> A potential negative societal impact is misuse through destructive spec in harmful MARL settings, we will add this to final version. In Tab.2 in the App., we discuss the computational overhead introduced by HyPOLE. We will make sure to mention these points in the final version.
>
> **Since the reviewer indicated `Good` for all evaluation criteria if our responses address the comments, we respectfully ask them to consider raising your score.**

---

> > ### Author Rebuttal · Reviewer_T79a · 2026-04-03
> >
> > Thanks for providing the MessySMAC results, which look more convincing to me and clearly strengthen the contribution. Together with the novelty that I highlighted earlier, I am now more confident to raise my score.

---

> > > ### Author Response · Authors · 2026-04-06
> > >
> > > We sincerely thank Reviewer T79a for acknowledging our rebuttal, increasing their score, and providing constructive feedback. We will include the new MessySMAC experiments in the final version to better strengthen our contribution, and we thank you for this suggestion for improving our paper.

---

### Official Review · Reviewer_yhGH · 2026-03-12

**Soundness:** 4
**Presentation:** 4
**Significance:** 4
**Originality:** 4
**Overall Recommendation:** 6
**Confidence:** 4

**Summary:**

The paper proposes a method to decompose multi-agent tasks expressed through HyperLTL to allow for training in a decentralized regiment to solve tasks that have previously required a centralized approach. The results are evaluated on a wide range of specifications and multiple environments.

**Compliance With Llm Reviewing Policy:**

Affirmed.

**Key Questions For Authors:**

1. In the paper you defined the POMDP with both finite sets for the states and actions. How would you approach extended to settings with either continuous states and/or actions?

**Limitations:**

I would like for the authors to elaborate on when your proposed method may not be the method of choice. What kind of problems/scenarios can a decentralized solution not solve?

**Strengths And Weaknesses:**

Soundness:

The approach proposed in the paper is well supported both by theoretical contributions through theorems, but also through well performed experiments on a wide range of tasks. The authors compare to a wide range of SOTA methods as well as in-depth ablations to prove the quality of their method. Nice work!

Presentation:

The paper is very well written and well motivated. The notation is clearly defined consistently throughout the paper. Given that it is a paper that relies on formal methods related approaches it is expected to be notation dense. I commend the authors for being meticulous and consistent with the notation. The only comment I have is that personally I would prefer to have the contents of the second paragraph in the introduction section come at a later stage where the notation $\forall\forall$ and $\forall\exists$ is already defined, as it is hard for the reader to grasp the meaning of this before the notation has been introduced. Consider rewriting this without the need of this notation in the introduction, or move it to a later part (perhaps closer to the problem statement where a connection and example to what these two different requirements may look like as part of the forest fire environment). It can also be left as it is without any major issues too.

I also came across some minor types that can easily be remedied:

1. Line 138: “and \psi is a inner […]” <- an
2. Line 208: “The updated statement is shown shown in Eq. (1)” <- Double shown


Also, Dec-POMDP is used once before it is defined from Decentralized-POMDP.
As a whole, these small mistakes do not take away from the general quality of the paper which is very well written.

Significance:

The paper is very well position and addresses a current problem in how to accurately train policies in multi-agent settings where policies need to satisfy predefined behavior prescribed by logic formulations. The paper is written with a general notation, applicable to many problem settings and allows for mappings to problems beyond the evaluations in the paper itself. Overall the paper is very well positioned and the problem setting is well motivated.

Originality:

The proposed method in the paper is original and non-trivial. It provides a substantial contribution for the SOTA and allows for a much needed relaxation in terms of communication needs between agents for a coordinated MARL setting with long-term dependence.

---

> ### Author Rebuttal · Authors · 2026-03-30
>
> We thank reviewer yhGH for the encouraging and kind feedback. We will make sure to address the comments about introducing the quantifiers in the introduction section. We will also fix the minor comments in the final version. Below, we address the reviewer’s questions.
>
> > Question: In the paper, you defined the POMDP with both finite sets for the states and actions. How would you approach extended to settings with either continuous states and/or actions?
>
> Thank you for the question. HyPOLE can be extended to continuous settings. This limitation mainly stems from our current use of **value-based CTDE methods**. To extend HyPOLE to continuous settings, one has to replace the current MARL backbone with **actor-critic CTDE methods** such as MADDPG or MATD3. This would also require adapting some definitions, in particular changing the policy class from deterministic to stochastic so that it is compatible with such methods. However, by changing from value-based CTDE to actor-critic CTDE methods, we will gain some advantages, such as support for continuous action spaces, but we may also **lose soundness**, since these methods do not provide an IGM-like structural guarantee connecting the centralized objective to decentralized optimal action selection. This property is essential for our Theorem 5.3, which is why our current theory is built on value-based CTDE methods. We will clarify this point in the final version.
>
>
> > Limitation: I would like for the authors to elaborate on when your proposed method may not be the method of choice. What kind of problems/scenarios can a decentralized solution not solve?
>
> HyPOLE is not the method of choice when the task does not involve **relational dependencies** or **temporal behaviors** between agents, and is instead well captured by standard reward design (sparse reward), such as tasks with largely independent objectives or simple non-temporal requirements. In such settings, the overhead of specifying HyperLTL formulas may not be justified. In addition, as the reviewer noted, the current version of our framework is instantiated with value-based CTDE methods, and therefore does not directly handle continuous action spaces. We will make sure to clearly raise these limitations in the final version.

---

> > ### Author Rebuttal · Reviewer_yhGH · 2026-04-03
> >
> > Thank you for answering the clarifying questions. My grade still remains and I believe that this paper is a great contribution to logic prescribed behavior for multi-agent coordination.

---

> > > ### Author Response · Authors · 2026-04-06
> > >
> > > We are very grateful to Reviewer yhGH for acknowledging our rebuttal and for their supportive feedback. We truly appreciate your encouragement. In the final revision, we will make sure to address the comment about introducing the quantifiers more clearly in the introduction.

---

### Official Review · Reviewer_rgPt · 2026-03-12

**Soundness:** 2
**Presentation:** 2
**Significance:** 1
**Originality:** 1
**Overall Recommendation:** 4
**Confidence:** 2

**Summary:**

This paper proposes HYPOLE, a MARL method that leverages HyperLTL specifications in POMDP environments. HYPOLE consists of three components: (1) HyperLTL Skolemization, (2) reward shaping based on a quantitative robustness function, and (3) RL training, all structured in accordance with the CTDE paradigm. Experiments are conducted in SMAC and Wildfire environments, and the results demonstrate the effectiveness of the proposed method through a variety of evaluations.

**Compliance With Llm Reviewing Policy:**

Affirmed.

**Final Justification:**

The authors provided a strong rebuttal that addressed all of my concerns with both theoretical arguments and concrete experimental evidence. The novelty over HypRL is better justified by clarifying the non-trivial challenges introduced by partial observability and decentralization. The additional Wildfire experiments convincingly demonstrate that the ∀∃ vs ∃∀ distinction generalizes beyond a single map, and the extended training results reveal substantial performance improvements on hard scenarios that were not apparent in the original submission. Given that all major weaknesses are satisfactorily resolved, I raise my recommendation to 4.

**Key Questions For Authors:**

Could you please address the weaknesses raised above?

**Limitations:**

yes

**Strengths And Weaknesses:**

**Strengths**

- This work extends the prior work HYPRL [1] from the MDP setting to the more practical POMDP setting, which is a meaningful direction in the field of formal specification-guided MARL applicable to many CTDE MARL algorithms.
- The effectiveness of the proposed method is demonstrated by applying it to various baseline algorithms across diverse environments.
- The paper shows that different HyperLTL formulas can lead to different learned strategies in the same environment, highlighting the importance of specification quality.

**Weaknesses**

- The overall pipeline is largely similar to that of HYPRL [1]. Rather than proposing a fundamentally new algorithm, the contribution appears to be the addition of CTDE and an encoder-decoder architecture, which limits the perceived novelty.
- The difference between $\forall \exists$ and $\forall \forall$ specifications is only demonstrated on a single map (MMM), which is insufficient to show that this difference generalizes. Furthermore, providing qualitative analysis of how the actual behaviors and trajectories differ under each specification would help readers better understand the effect.
- The performance gap between the baselines and the proposed method does not appear to be substantial in the experimental results.

**Minor Comments**

- Figure 1 is too small. Given the complexity of the architecture, enlarging the figure would significantly help readability and understanding.

[1] Hsu, T.-H., Rafieioskouei, A., and Bonakdarpour, B. HYPRL: Reinforcement learning of control policies for hyperproperties. NeurIPS 2025.

---

> ### Author Rebuttal · Authors · 2026-03-30
>
> We thank reviewer rgPt for their detailed review. Below, we address each comment.
>
> >[W1] Overall... HYPRL...perceived novelty.
>
> Thanks for the comment, but we respectfully disagree. While HyPOLE builds on ideas from HypRL (e.g. learning Skolem functions and the quantitative semantics), it introduces **substantial and non-trivial advances** due to **partial observability** and **decentralization**, which fundamentally change both the formulation and the learning setting. Moreover, [ICML Guideline](https://tinyurl.com/mv5rf538) do not require an entirely new method for a contribution to be considered original. Below, we elaborate further:
> **Theoretical perspective:** Sec.5.3 provides details of lifting POMDPs to Dec-POMDPs to allow decentralized control and formally shows that policies learned via CTDE correspond to the optimal policies in our problem setting (Fig.3). This is is fundamentally different from HypRL and far from a simple extension. In addition, in our problem statement and in Sec.5.1 and 5.2, we extend Skolemization and Quantitative Semantics to the POMDP setting (i.e., sampling and optimization have to consider observation histories instead of state traces), which is also non-trivial.
> **Practical perspective:** HypRL is limited to fully observable, centralized settings, whereas HyPOLE addresses decentralized settings under partial observability, which is a significantly more realistic regime. This shift leads to a different problem, including a clear distinction between $\forall\forall$ and $\forall\exists$, which HyPOLE captures through separate pipelines. In particular, under $\forall\exists$ in the decentralized setting, a Skolem function cannot be computed (due to partial observability) and one has to estimate the trajectories of the preceding $\forall$-agents through the seq2seq model without access to global states. In contrast, in HypRL, both policies and the Skolem functions are learned by a centralized component. We also argue that in a centralized setting, these two quantifier structures often lead to similar outcomes due to access to global states.
> In summary, Dec-POMDPs subsume MDPs (Fig.1 in [Link](https://tinyurl.com/28cb2hee)), meaning that HyPOLE generalizes HypRL. If accepted, we will make these important distinctions clearer.
>
> >[W2] Difference $\forall\exists$...understand effect.
>
> In the submission, we evaluated the difference between $\forall\exists$ and $\forall\forall$ on two SMAC settings, including MMM (Fig.6) and additional plot on MMM2 (Fig.11 in App). We will reference it in Sec.6. To further show that $\forall\exists$ more effectively guides the search towards optimal policies, during the rebuttal period, we conducted additional experiments on two maps in the wildfire scenario (Fig.3 in [Link](https://tinyurl.com/28cb2hee)). In both maps, VDN+HyPOLE($\forall\exists$)(purple) converges faster than VDN+HyPOLE($\forall\forall$)(orange), and on the 8P_3F2V map this improvement is substantial. We also observe improvements for QTRAN+HyPOLE($\forall\exists$)(brown) compared to QTRAN+HyPOLE($\forall\forall$)(green) on both maps, especially on 8P_3F2V.
>
> In addition, we provided a qualitative illustration in the example (line 144-164 right and App.B), and show, changing the quantifiers from $\forall\exists$ to $\forall\forall$ reduces the probability of satisfaction from 1 to 0.75, as $\forall\forall$ is a stronger requirement. To improve clarity, if accepted, we'll use extra page to move App.B into paper and better highlight how behaviors differ under each specification.
>
> >[W3] Performance... substantial...results.
>
> HyPOLE achieves **substantial improvements** compared to baselines. For this rebuttal, we extended corridor to 5M steps (Fig.4 in [Link](https://tinyurl.com/28cb2hee), where QMIX+HyPOLE reaches up to 60% win rate while QMIX remains negligible. In (see Fig.5 [Link](https://tinyurl.com/28cb2hee)), we provide plots demonstrating the gains across all maps. QMIX+HyPOLE (blue) outperforms QMIX on almost all maps except 5m_vs_6m and 3s5z, where in 3s5z it still surpasses in the final steps. In hard scenarios such as MMM2 (Fig.5g), bane_vs_bane (Fig.5h), and corridor (Fig.5i), QMIX+HyPOLE shows **significant improvements**, with **up to 90% gain** in success rate on bane_vs_bane. Similarly, VDN+HyPOLE (orange) improves performance in most cases, except 2s3z where the behavior is more unstable, while still achieving **substantial gains** on bane_vs_bane and. Finally, QTRAN+HyPOLE (green) outperforms the baseline in nearly all scenarios except 5m_vs_6m, and achieves **notable improvements** on MMM with **up to 50% gain** (Fig.5f). In addition to CTDE methods, IQL+HyPOLE (red) achieves good results on almost all maps, specifically **up to 60% gain** on 8m (Fig.5d).
>
> >Fig.1 is small.
> We agree, if the paper gets accepted, we will use the extra page to enlarge fig.1.
>
> **If our responses have addressed your comments, we respectfully ask you to consider raising your score.**

---

> > ### Author Rebuttal · Reviewer_rgPt · 2026-04-04
> >
> > I thank the authors for the thorough rebuttal with concrete experimental evidence. My concerns are adequately addressed: the novelty over HypRL is better justified through the non-trivial challenges of partial observability and decentralized Skolemization, the ∀∃ vs ∃∀ difference is now demonstrated across multiple environments including Wildfire, and the extended experiments clearly show substantial performance gains especially on hard maps. I raise my score to 4.

---

> > > ### Author Response · Authors · 2026-04-06
> > >
> > > We sincerely thank Reviewer rgPt for acknowledging our rebuttal and for increasing their score. In the final revision, we will enlarge Fig. 1 and include the new graphs comparing $\forall\forall$ and $\forall\exists$, and revise Section 6 to better compare the performance gap between HyPOLE and the baseline. We thank Reviewer rgPt for these constructive suggestions.

---

### Decision · Program_Chairs · 2026-04-30

**Decision:**

Accept (regular)

**Comment:**

The article caused a very interesting discussion among the reviewers. The main question is whether the proposed approach is sufficiently new in relation to similar work HypRL for a one-agent scenario. Among the arguments for the lack of significance of the work was that the transfer of LTL logic to the QMIX algorithm is generally straightforward, and the notation ∀∀ vs ∀∃ has already been encountered in other simpler multi-agent problems. Nevertheless, in my opinion, the work done by the authors to adapt the pipeline from the work of HypRL is significant. The work is technically valid and no reviewer has noted any significant flaws in the theorems or experiments. The article is well written and well positioned relative to other work, in particular HypRL. I am sure that the article will be useful to the MARL community and it can be accepted to the conference.